# Assessing geographic controls of hair isotopic variability in human populations: A case-study in Canada

Clement P. Bataille ⓘ *, Michelle M. G. Chartrand, Francis Raposo, Gilles St-Jean

Department of Earth and Environmental Sciences, University of Ottawa, Ottawa, Ontario, Canada

* cbataill@uottawa.ca

**Data Availability Statement:** The isotope data, food frequency questionnaire and R script required for this paper are available in supplementary material.

## Abstract

Studying the isotope variability in fast-growing human tissues (e.g., hair, nails) is a powerful tool to investigate human nutrition. However, interpreting the controls of this isotopic variability at the population scale is often challenging as multiple factors can superimpose on the isotopic signals of a current population. Here, we analyse carbon, nitrogen, and sulphur isotopes in hair from 590 Canadian resident volunteers along with demographics, dietary and geographic information about each participant. We use a series of machine-learning regressions to demonstrate that the isotopic values in Canadian residents' hair are not only influenced by dietary choices but by geographic controls. First, we show that isotopic values in Canadian residents' hair have a limited range of variability consistent with the homogenization of Canadian dietary habits (as in other industrialized countries). As expected, some of the isotopic variability within the population correlates with recorded individual dietary choices. More interestingly, some regional spatial patterns emerge from carbon and sulphur isotope variations. The high carbon isotope composition of the hair of eastern Canadians relative to that of western Canadians correlates with the dominance of corn in the eastern Canadian food-industry. The gradient of sulphur isotope composition in Canadian hair from coast to inland regions correlates with the increasing soil pH and decreasing deposition of marine-derived sulphate aerosols in local food systems. We conclude that part of the isotopic variability found in the hair of Canadian residents reflects the isotopic signature associated with specific environmental conditions and agricultural practices of regional food systems transmitted to humans through the high consumption rate of intra-provincial food in Canada. Our study also underscores the strong potential of sulphur isotopes as tracers of human and food provenance.

## Introduction

Isotopes of elements inherited from dietary sources such as carbon (C), nitrogen (N), and sulphur (S) have emerged as powerful tools to study human nutrition [1,2]. In recent years, several studies have used the C, N and S isotope composition in human tissues ($\delta^{13}$C, $\delta^{15}$N, and $\delta^{34}$S) to investigate differences in human nutrition and dietary choices at the population scale

**Funding:** Funding: C.P.B. and F.R. acknowledge funding from Canadian Security and Safety Program Targeted Investment (CSSP-2018-TI-2385). G.S.J and M.M.G.C. acknowledge funding from the Chemical, Biological, Radiological and Nuclear Research & Technology Initiative (CRTI 08-0116RD). Author contributions: C.P.B. G.S.J and M.M.G.C designed the project and analyzed the data sets. C.P.B. and F.R. performed the statistical analysis and model development steps. All authors contributed to the interpretation of the results and writing of the manuscript. Competing interests: The authors declare that they have no competing interests. Data and materials availability: All data needed to evaluate the conclusions in the paper are present in the paper and/or the Supplementary Materials. Additional data related to this paper may be requested from the authors.

**Competing interests:** The authors have declared that no competing interests exist.

[Reviewed in 3]. Keratinized tissues, hairs and nails, are an ideal substrate for analyzing $\delta^{13}$C, $\delta^{15}$N and $\delta^{34}$S values in dietary studies, as these elements are abundant in keratin and record dietary isotopic values chronologically as they grow [4–7]. Isotope data from hair can thus provide snapshots of the diet from an individual or population at a monthly temporal resolution [8–12]. Stable isotope variations in hair have been particularly successful in demonstrating dietary transition in human populations by tracing, for example, the progressive increase in processed food products in certain indigenous community diets [13–18]. The comparison of isotope hair compositions between countries also gives some information about dietary homogenization at the global scale [5,19–23]. In many countries, $\delta^{13}$C, $\delta^{15}$N and $\delta^{34}$S composition in human hair or nails are becoming increasingly homogeneous due to the globalisation of food trade and the homogenization of dietary habits, particularly in urban regions [20]. On the other hand, regional dietary traditions (e.g., high rate of seafood consumption) contribute to a higher isotopic variability within and between populations in less industrialized regions [13].

However, even when food products are chemically identical, their isotopic composition can differ depending on the environmental, agricultural or manufacturing conditions during their production [24]. As global food items are often produced using local agricultural products (i.e., "glocalization"), food systems can inherit distinct isotopic values depending on their geographic origin [24]. Consequently, the preferential consumption of glocal food by certain populations could add some isotopic variability at regional to global scales that not only reflect individual dietary choices but vary geographically with the local food isotopic baselines [19]. For example, $\delta^{13}$C values in human hair increase in populations living closer to the equator, correlating the higher proportion of isotopically heavy $C_4$ crops in those regions [25]. Even with identical diets, individuals from Brazil and from Europe will display distinct $\delta^{13}$C values in their tissues because isotopically heavy $C_4$ crop by-products are used throughout the food industry in Brazil (e.g., cattle feed, sugar) whereas isotopically light $C_3$ plants are dominant in Europe [26]. Many other agricultural or environmental factors can influence the isotopic baselines of food consumed by human populations at the regional scale, overprinting the influence of individual dietary choices and complicating interpretations.

$\delta^{13}$C, $\delta^{15}$N, and $\delta^{34}$S in human populations can vary with multiple processes, including metabolic fractionation, mixture of isotopically distinct food items, and variability in local food isotopic baselines. However, disentangling these factors for each isotopic system is often challenging. $\delta^{15}$N values increase moving up the trophic chain because N isotopes fractionate during N excretion [22,27–31]. Consequently, humans eating more animal protein, particularly seafood, tend to have higher $\delta^{15}$N values than vegetarians [29]. Conversely, humans eating more legumes tend to have lower $\delta^{15}$N values [32]. $\delta^{15}$N values can also transiently be affected by metabolic fractionation associated with physiological stress (e.g. anorexia, bulimia, pregnancy or certain diseases) [33]. However, $\delta^{15}$N baselines in food can also vary with environmental and agricultural practices at the site of food production, including climate [34,35], soil properties [36], or fertilization practices [37]. Metabolism and trophic level play only minor roles in determining $\delta^{13}$C values in human tissues [30,38]. Most of $\delta^{13}$C variability in modern humans usually reflect the proportion of $C_4$ vs. $C_3$ in the food consumed [3,29,39] because $C_3$ plants (e.g., wheat, rice, sugar beets) have much lower $\delta^{13}$C values ~-25‰ than $C_4$ plants (e.g., corn, sugar cane) ~-12‰ [40]. Livestock fed on a $C_4$ diet (e.g., corn) has higher $\delta^{13}$C values than cattle fed a $C_3$ diet (e.g., barley) and those isotopic differences are further propagated to human tissues based on their dietary sources (e.g., meat, sugar) [24,27]. Seafood has also a distinct $\delta^{13}$C value generally between -17‰ and -20‰, which can influence the $\delta^{13}$C value of population with high seafood consumption [15]. Additionally, $\delta^{13}$C baselines in food can also vary with environmental and agricultural

practices at the site of food production including climate [41], soil properties [41], fertilization practices [42], elevation [43], or even $CO_2$ levels and aerosol deposition [42]. $\delta^{34}S$ values in human tissues reflect in part the amount of seafood consumed [44] because the oceans have a high and isotopically uniform $\delta^{34}S$ transmitted to marine food chains [45]. Though results remain ambiguous, $\delta^{34}S$ values are potentially influenced by trophic level [30,46,47] and internal metabolism [48]. $\delta^{34}S$ baselines in food systems also display spatial patterns related to the local environmental conditions at the site of production [49] including bedrock geology [50,51], climate [52], soil properties [52], aerosol deposition [50] and fertilization practices [53,54].

We attempt to disentangle the controls of isotopic variability in human hair at the population scale through a case study in Canada. We focus particularly on assessing the variability of isotopic baselines in food systems. Canada is known to be a strong agricultural nation, with major agricultural centers throughout the country. Those agricultural centers produce the majority (>70%) of the food that Canadians consume [55]. The Canadian agrifood business is well-integrated at the provincial level and favors strong intra-provincial consumption of the produced food [56]. Besides direct retailing by producers, Canadian glocal food includes retailing of processed food generated from locally produced and distributed agricultural goods [56]. Canada's large size also favors strong intra-provincial agricultural markets because it is often less economical to transport agricultural goods between provinces [55]. Consequently, a large portion of the Canadian residents' diet is glocal [55]. Because Canada is a large country with vastly different environmental conditions and crop distribution throughout its territories, this diversity of environmental and climatic conditions of agricultural zones should lead to distinct isotopic baselines in glocal food produced throughout the country. We hypothesize that if the food consumed by Canadians is dominantly glocal in origin, tissues of Canadian residents should have distinct isotopic values between provinces/regions across Canada reflecting the isotopic baseline of regional food systems. One possible method to assess these geographic controls on isotopic variability in hair of Canadian residents would have been to collect common food items (e.g., meat, eggs, milk, cereals) from different Canadian regions to explore possible spatial trends. However, we would have had to collect many different food items to represent properly the integrated Canadian diet. To limit cost and time, we explored the presence of spatial trends directly from the isotopic variability in hair of Canadian volunteers using multivariate regression. We collected $\delta^{13}C$, $\delta^{15}N$ and $\delta^{34}S$ values in human hair from local volunteers across Canada including resampling of several volunteers through a 4-year period. We combined those isotopic data with demographics, and dietary habits of volunteers as well as environmental/agricultural conditions at their residence location. We then used a series of random forest regressions to sequentially assess how dietary choices, demographics and geography influence isotopic variability.

## Materials and methods

### Ethical statement and data availability

The research procedure was approved by the Office of Research Ethics and Integrity of the University of Ottawa (Ethics File number: H10-17-10). Specifically, all sampling and analytical methods used were in accordance with relevant guidelines from the Office of Research Ethics and Integrity of the University of Ottawa. Written consents were obtained from all participants in accordance with and maintained under regulations from the Office of Research Ethics and Integrity of the University of Ottawa. The isotope data, sample locations and compiled responses to FFQ are available in S1 and S2 Data.

## Participant recruitment, questionnaire and hair collection procedure

Between 2008 and 2012, poster announcements and mailings were sent to the Royal Canadian Mounted Police across the country and to personal contacts in target locations across Canada to recruit volunteers for an isotope and dietary survey. Among the respondents, 590 participants aged over 18 years old were selected for the study based on two criteria: 1) limited travel within the last year, and 2) coverage of the most populous regions across Canada. Except for age restrictions, we did not select participants based on demographics or socio-economic status. However, based on the study design and announcements, it is likely that some biases exist in the selected population. At entry into the study, participants answered a dietary, travel, and demographic questionnaire (S1 Table). Hair samples were then collected in two manners: haircuts (for those who have short hair), and cuts from the scalp (for those who had longer hair). For participants with long hair, only the hair part grown in the last year (i.e. the top 12 cm of hair from the scalp) was used in the analysis. Based on the travel questions, confidence is high that for most participants, the collected hairs were grown at the location of residence of the volunteer. However, about 15% of the participants traveled to a distant location within 1 year of hair collection. For those who had traveled, the length of hair associated to the time the participant travelled was estimated (assuming a constant hair growth rate of 1 cm/month), and a 3 cm segment of hair above the estimated length was removed from the sample, and not used in the analysis. This assumption adds some uncertainty to our analysis, as at any given moment at least 10% of hairs are not growing (resting phase) and hairs of different participant do not grow at the same rate [57]. Out of the 590 participants, 25 participants were resampled every 4 to 6 months (157 samples) for a period of 4 years. At each sampling period, they filled an additional questionnaire to assess any change in their dietary inputs and travel history. For logistical reasons, these resampling activities were performed in only 3 urban regions: Montreal (n = 4), Ottawa (n = 10) and Sudbury (n = 11).

## Isotopic analysis

Human hair is a particularly interesting substrate to investigate human nutrition, as the great majority of C, N, and S in hair keratin is derived from the food consumed by the consumer[3]. Hair is also rich in C, N and S, and is easily collected, is resistant, and does not undergo elemental turnover [4]. All hair for isotope analysis was prepared by first washing the hair in a 2:1 solution of chloroform:methanol ($CHCl_3$:MeOH), drying the hair, then grinding the hairs into a powder using a Retsch ball mill and stainless steel grinding jars. The hair was then stored in glass vials until analyzed. Hair samples were analyzed for $\delta^2H$, $\delta^{13}C$, $\delta^{15}N$, $\delta^{34}S$, $\delta^{18}O$, and $^{87}Sr/^{86}Sr$ values. However, in some cases, there was not enough hair for all isotopic analyses. In this work, only the $\delta^{13}C$, $\delta^{15}N$, and $\delta^{34}S$ data are reported. For $\delta^{13}C$ and $\delta^{15}N$ analysis, the samples and isotope standards were weighted into tin capsules and loaded onto an Elemental Analyser (Vario EL cube, Elementar, Germany) interfaced (Conflow III, Thermo, Germany) to an isotope ratio mass spectrometer (IRMS, Delta$^{Plus}$ Advantage, Thermo, Germany). Internal standards used for calibration were a mix of ammonium sulfate and sucrose ($\delta^{13}C_{VPDB}$, -11.94‰; $\delta^{15}N_{AIR}$, 16.58‰), nicotinamide ($\delta^{13}C_{VPDB}$, -22.95‰; $\delta^{15}N_{AIR}$, 0.07‰), and caffeine ($\delta^{13}C_{VPDB}$, -28.53‰; $\delta^{15}N_{AIR}$, -3.98‰). All $\delta^{15}N$ and $\delta^{13}C$ values are reported versus AIR and VPDB, respectively. Analytical precision was based on an internal quality check reference sample (glutamic acid, which is not used for normalization) and was better than ± 0.2 ‰ for both $\delta^{13}C$ and $\delta^{15}N$. For $\delta^{34}S$ analyses, the samples and standards were weighted into tin capsules, loaded onto an Isotope cube Elemental Analyser, and flash combusted at 1800°C. The EA method was optimized for $SO_2$; $N_2$ and $CO_2$ were not retained. $SO_2$ was trapped and released to the Conflo IV (Thermo, Germany) interfaced to the IRMS (DeltaPlus XP, with special 6

collector sulphur cups (SO-SO$_2$), Thermo, Germany). Standards used for calibration were silver sulphides: IAEA-S-1 ($\delta^{34}$S, -0.3‰), IAEA-S-2 ($\delta^{34}$S, 22.7‰), and an internal standard S-6 ($\delta^{34}$S, -0.7‰). All $\delta^{34}$S values were reported to the international scale VCDT. Four ground human hair samples were used as quality check [58]. Analytical precision is based on the reproducibility of the in-house hair standards AND (G737), COL (G738), CAL-CAN (G739) and CAL-SAL (G740) which was better than ± 0.3‰.

## Predictors and machine-learning regression

All statistical analyses and figures from this manuscript were conducted in R programming language version Rx64 3 4.2. (https://www.r-project.org/). An example of R-script is available in S4 R Script. We incrementally tested if dietary choices and demographics of volunteers (*Step 1*), location of residence (*Step 2*) and environmental conditions at the site of residence (*Step 3*) were important predictors of isotopic variability in hair of Canadian residents. For each step, we first used the Pearson correlation coefficient and anova, independent t-test and Levene's test to assess if significant relationships existed between isotope data and continuous and categorical predictors, respectively.

We then integrated categorical and continuous predictors into a random forest multivariate regression. We choose random forest over generalized linear model and other machine-learning algorithms (e.g., support vector machine) because it requires very little pre-processing and can integrate the categorical (e.g., province, sex, age) and numerical variables (e.g., latitude, longitude, mean annual temperature) of our dataset into the same regression framework [59]. Random forest is a flexible and interpretable tree-based machine-learning algorithm trained by bootstrap sampling and random feature selection. Random forest creates multiple decision trees on different data samples where sampling is done with replacement to prevent overfitting. To make fair use of all potential predictors, the number of features split at each node of a tree is limited to some user-defined threshold. Ultimately, random forest aggregatesthe results of these decision trees to predict the mean value of the response variable, in our case the isotopic composition of hair. To maximize model performance while minimizing the number of predictors included, we used the Variable Selection Under Random Forest (*VSURF*) package, which helps eliminate irrelevant and redundant variables [60]. *VSURF* uses a two step-process, first ranking variables and then selectively adding variables into a model to minimize out-of-bag error. A series of random forest regression models were developed for each isotopic system incrementally testing the potential of different predictors to explain isotopic variability (Fig 1). We compared the performance of each of these models to determine which of the variables could explain most of the variance for each isotopic system.

*In step 1*, we tested the predictive power of dietary choices and demographic using data from the questionnaire (S1 Table). The dietary variables included consumption amount of different beverages in milliliters per week (i.e., bottled water, milk, soda, wine, beer, coffee, and other beverages), presence of meat in the diet (grouped in 2 categories—ovo-lacto vegetarian and meat consumer), and the amount and type of seafood consumed (S1 Data). The demographic variables included sex (male and female), age (grouped in 6 categories), and smoking habits (smoker and non-smoker) (S1 Data). We tested if significant relationships existed between these dietary choices and demographic variables and each isotopic system. We then used the selected significant predictors and *VSURF* to develop a random forest regression model for each isotopic system (Regression 1 in Fig 1).

*In step 2*, we tested the predictive power of geographic predictors including latitude, longitude and province of residence of the volunteers obtained during field collection (S1 Data). Prior

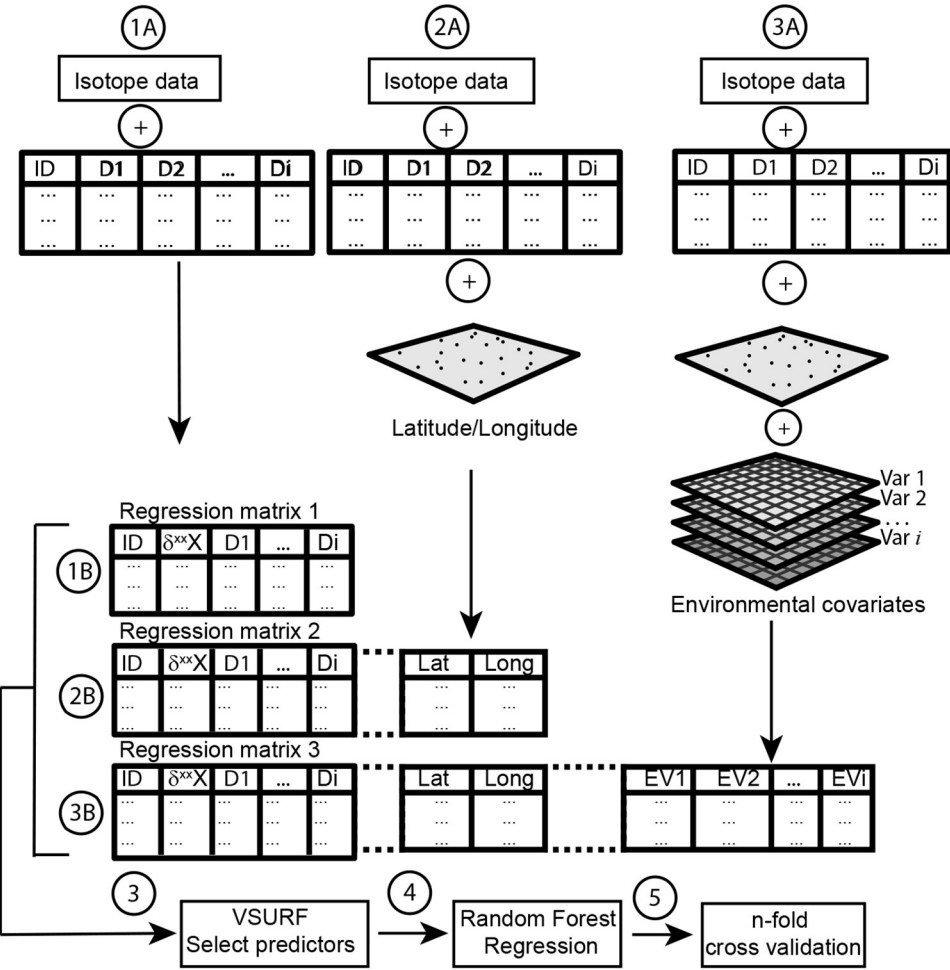

**Fig 1. Multiple machine-learning regression workflow (see Materials and methods).** Regression 1 includes only isotope data, dietary choices, demographic variables (S1 Table); regression 2 includes isotope data, dietary choices, demographic variables, and latitude/longitude; and regression 3 includes isotope data, dietary choices, demographic variables, and environmental covariates (Table 1).

to testing correlations, we tested the presence of spatial autocorrelation in each isotopic dataset by calculating semivariograms (Fig 4). Semivariograms represent how semivariance changes as the distance between observations changes. A constant semivariance indicates not spatial autocorrelation, whereas an increasing semivariance indicates some spatial autocorrelation. We then tested if significant relationships existed between these geographic variables and each isotopic system. Last, we used the selected significant geographic, dietary choices and demographics predictors within *VSURF* to develop a random forest regression model for each isotopic system (Regression 2 in Fig 1).

*In step 3*, we tested the predictive power of environmental conditions at the site of residence. We used the latitude and longitude of each collection site to extract local environmental conditions at the site of residence using open-access geospatial data (Table 1). We assumed that the local environmental conditions at the site of residence were a good approximation of the local environmental conditions of local food systems. To assess these local environmental conditions of food systems, we selected geospatial data representing known factors of isotopic variability (Table 1). We resampled and reprojected all the selected

**Table 1. Auxiliary variables.** List of geological, climatic, environmental and topographic variables used for the regression.

| | Variables | Initial resolution | Source |
|---|---|---|---|
| Variable | Elevation | 90 m | [61] |
| Clay | Surficial Soil Clay (Weight%) | 250 m | [62] |
| Ph | Soil pH in $H_2O$ solution | 250 m | [62] |
| MAT | Mean Annual Temperature (WorldClim) | 30-arc sec | [63] |
| MAP | Mean Annual Precipitation (WorldClim) | 30-arc sec | [63] |
| PET | Global potential evapotranspiration | 30-arc sec | [64] |
| Elevation | Global elevation dataset | 30-arc sec | [65] |
| SUL | Sulphur Deposition (CMIP3 NINT simulation) | 2.5-degrees | [66] |
| BCB | Biomass Black Carbon (CMIP3 NINT simulation) | 2.5-degrees | [66] |
| BCA | Industrial Black Carbon (CMIP3 NINTsimulation) | 2.5-degrees | [66] |
| AOD | Aerosol Optical Depth (Particles < 2.5microns) | 0.1-degrees | [67] |
| Salt | Sea Salt Deposition (CCSM.3 Simulation) | 1.4-degrees | [68] |
| Dust | Dust deposition (Multi-model average) | 1.0-degrees | [68] |
| Nfert | Global Fertilizer, Version 1 | 0.5 degrees | [69] |
| Nman | Global Manure, Version 1 | 0.5 degrees | [69] |
| Corn | Distance to major corn producing center in Canada* | 1 km | This study |
| sugar | Distance and type of sugar refineries** | 1 km | This study |

\*\* Calculated using the distance tool in ArcGIS and centroids from the two major corn producing centers in Canada: Southern Ontario and Saint Lawrence River valley

\*\* Calculated using the Inverse Distance Weighing tool in ArcGIS and locations of the 5 Canadian sugar refineries with 0 = beat sugar refinery and 1 = sugarcane refiner

environmental geospatial products into WGS84-Eckert IV 1km$^2$ resolution and used latitude and longitude to extract the local values for each raster. For nitrogen isotopes, the selected auxiliary variables include climates (e.g., temperature and mean annual precipitation) [34,35], soil properties (e.g., clay content and organic matter content) [36], and fertilization practices (e.g., synthetic vs. manure fertilizers) [37]. For carbon isotopes, the selected auxiliary variables include C3 vs. C4 crop distribution (e.g., distance and type of grain mill, distance, and type of sugar refineries) [40], climate (e.g., temperature and mean annual precipitation) [41], soil properties (e.g., pH, clay content, organic matter content) [41], elevation [43], $CO_2$ levels and anthropogenic aerosol deposition [42], and fertilization practices [42]. For sulphur isotopes, the variables include bedrock geology (e.g., rock type) [50], climate (e.g., precipitation, temperature) [52], soil properties (e.g., pH, clay content, organic matter) [52] aerosol deposition (e.g., sea salt, anthropogenic) [50] and fertilization practices [53]. We then tested if significant relationships existed between these environmental variables and each isotopic system (Fig 3). Last, we used the selected significant environmental, geographic, dietary choice and demographics predictors within *VSURF* to develop a random forest regression model for each isotopic system (Regression 3 in Fig 1).

## Results

### Isotopic data in Canadian hair

We analyzed a total of 577 hair samples for $\delta^{13}C$ and $\delta^{15}N$ values and 549 hair samples for $\delta^{34}S$ values from participants across Canada (S1 Data and Fig 2).

The $\delta^{15}N$, $\delta^{13}C$ and $\delta^{34}S$ values in Canadian hair ($\delta^{15}N_{hair}$, $\delta^{13}C_{hair}$ and $\delta^{34}S_{hair}$) range from 7.6 to 10.8 ‰, -20.3 to -16.7 ‰, and -1.4 to 4.8 ‰, respectively (Fig 2). $\delta^{15}N_{hair}$, $\delta^{13}C_{hair}$, and

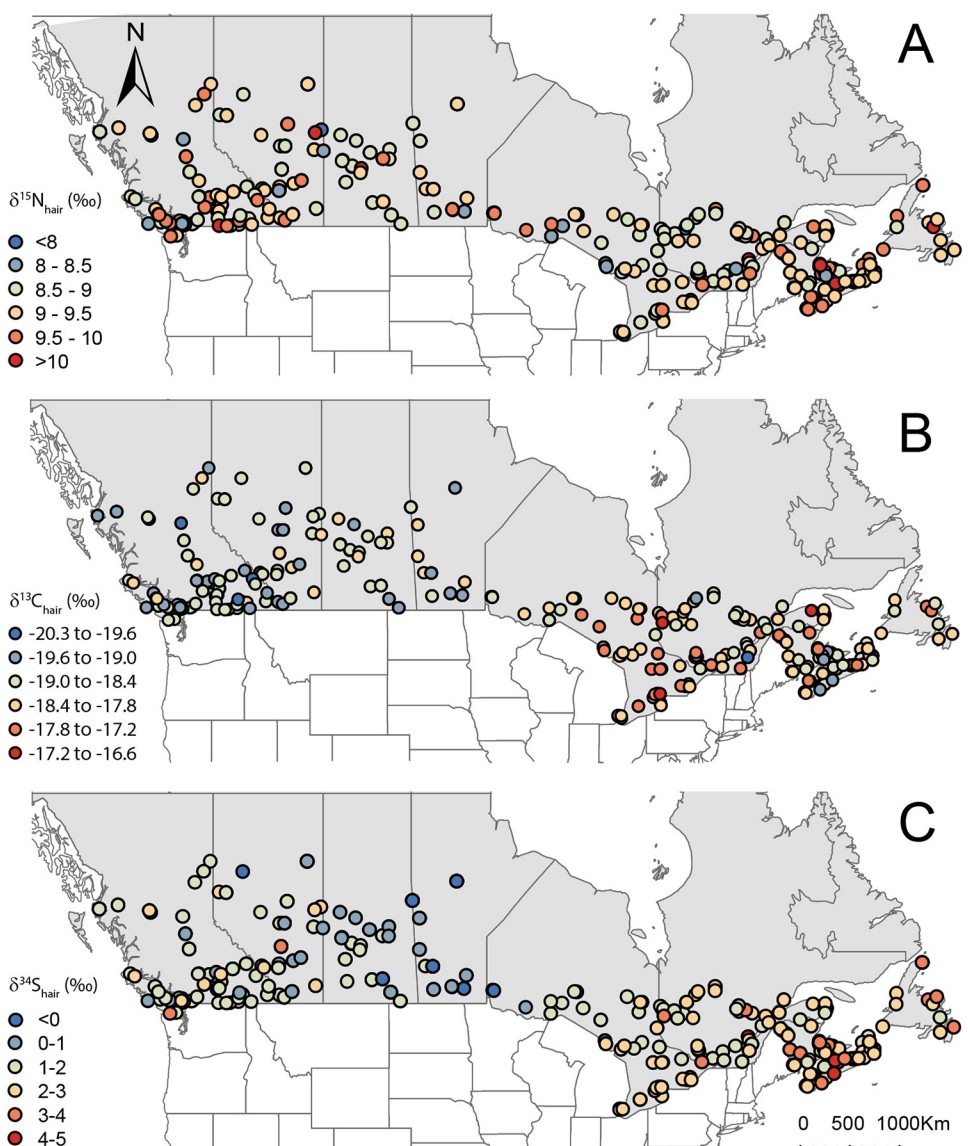

**Fig 2. Distribution of sample locations and isotopic values in hair across Canada.** A. Nitrogen isotope variations (n = 577). B. Carbon isotope variations (n = 577). C. Sulphur isotope variations (n = 549). Administrative boundaries are from http://www.naturalearthdata.com/. This map was generated in Rx64 3 4.2 (https://www.r-project.org/).

$\delta^{34}S_{hair}$ distribution are normally distributed (Shapiro test, p-value>0.05 for all three variables) (S2 Table). There is a weak but significant correlation between all isotope data (Pearson correlation; p-value<0.01).

## Isotopic data in Canadian hair compared to other countries

We compared isotopic data from Canadian hair to those collected from other studies (Table 2). We find that Canada has isotopic variability similar to other industrialized nations (e.g., Europe, USA) but lower than less industrialized countries. The average $\delta^{15}N_{hair}$ value is very similar to those observed in Europe, or the USA (Table 2). The average $\delta^{13}C_{hair}$ value falls

**Table 2. Comparison of $\delta^{15}N_{hair}$, $\delta^{13}C_{hair}$ and $\delta^{34}S_{hair}$ values between administrative regions or countries.** Data of $\delta^{15}N_{hair}$ and $\delta^{13}C_{hair}$ values exist for other countries but do not include $\delta^{34}S_{hair}$ values or only include a few individuals (for a summary or country level details about those isotopic data see [25] and [70]).

| | $\delta^{15}N_{hair}$ | $\delta^{13}C_{hair}$ | $\delta^{34}S_{hair}$ | Sample size | Reference |
|---|---|---|---|---|---|
| Canada | 9.2 ± 0.5 | -18.5 ± 0.6 | 1.7 ± 1.0 | 590 | This study |
| | 8.3 ± 0.5 | -18.2 ± 0.5 | 4.6 ± 1.3 | 15 | [70] |
| USA | 8.9 ± 0.4 | -17.2 ± 0.8 | 3.4 ± 1.1 | 234 | [20] |
| Europe | 9.2 ± 0.5 | -20.3 ± 0.8 | 6.9 ± 0.9 | 129 | [19] |
| | 8.6 ± 0.6 | -20.9± 0.5 | 6.7 ± 1.1 | 420 | [70] |
| Asia | 8.5 ± 1.3 | -20.0 ± 0.9 | 7.1 ± 1.5 | 137 | [70] |
| | 8.3± 1.3 | -20.1 ± 1.3 | 7.3 ± 1.7 | 144 | [21] |
| South America | 9.0 ± 0.7 | -17.8 ± 1.6 | 7.2 ± 2.5 | 76 | [70] |

between that of Europe and the USA (Table 2). The average $\delta^{34}S_{hair}$ value is lower than in other regions of the world (Table 2).

## Predictors and machine-learning regression models

**Dietary choices and demographic covariates.** $\delta^{15}N_{hair}$ values correlate positively with the amount of seafood consumed by volunteers (Table 3). $\delta^{15}N_{hair}$ values are significantly higher in participants eating meat than in ovo-lacto vegetarians and pescatarians (Table 4). Our most accurate random forest regression model to predict $\delta^{15}N_{hair}$ variations using dietary choices

**Table 3. Anova between isotopic data in hair and the categorical predictors.** Grey indicates significant differences in the mean isotopic values between groups of that variable (p-value<0.01). For those significant variables relation between groups and isotopic values are further tested through a t-test.

| Variables | $\delta^{15}N_{hair}$ | | $\delta^{13}C_{hair}$ | | $\delta^{34}S_{hair}$ | |
|---|---|---|---|---|---|---|
| | F value | p-value | F-value | p-value | F-value | p-value |
| Province | 0.7 | 0.4 | 6.8 | 0.009 | 39.2 | 1.5e-9 |
| Sex | 2.0 | 0.2 | 13.3 | 0.0003 | 4.1 | 0.04 |
| Age | 0.03 | 0.9 | 22.8 | 2.9e-6 | 0.1 | 0.7 |
| Hair dye | 2.3 | 0.1 | 9.5 | 0.002 | 1.3 | 0.3 |
| Water type | 2.8 | 0.1 | 1.5 | 0.2 | 1.3 | 0.3 |
| Vegetarian | 18.6 | 2.3e-6 | 8.5 | 0.004 | 0.6 | 0.4 |
| Smoker | 2.1 | 0.2 | 0.7 | 0.4 | 2.8 | 0.09 |

**Table 4. $\delta^{13}C_{hair}$, $\delta^{15}N_{hair}$ and $\delta^{34}S_{hair}$ values relative to meat consumption.** p-values from t-tests comparing $\delta^{13}C_{hair}$, $\delta^{15}N_{hair}$ and $\delta^{34}S_{hair}$ values between omnivores vs. pescatarians and ovo-lacto vegetarians. Pescatarians and ovo-lacto vegetarians were combined for the t-test because of the limited number of participants reporting this diet style.

| Group | Average $\delta^{13}C$ ± SD (‰) | Average $\delta^{15}N$ ± SD (‰) | Average $\delta^{34}S$ ± SD (‰) |
|---|---|---|---|
| Meat eater (n = 565) | -18.5 ± 0.6 | 9.2 ± 0.5 | 1.7 ± 1.0 |
| Pescatarian (n = 5) | -18.9 ± 0.9 | 8.9 ± 0.2 | 2.7 ± 1.3 |
| Ovo-lacto vegetarian (n = 7) | -19.2 ± 0.3 | 8.2 ± 0.5 | 1.8 ± 0.9 |
| t-test p-value | 0.004 | 1.5e-6 | 0.1 |

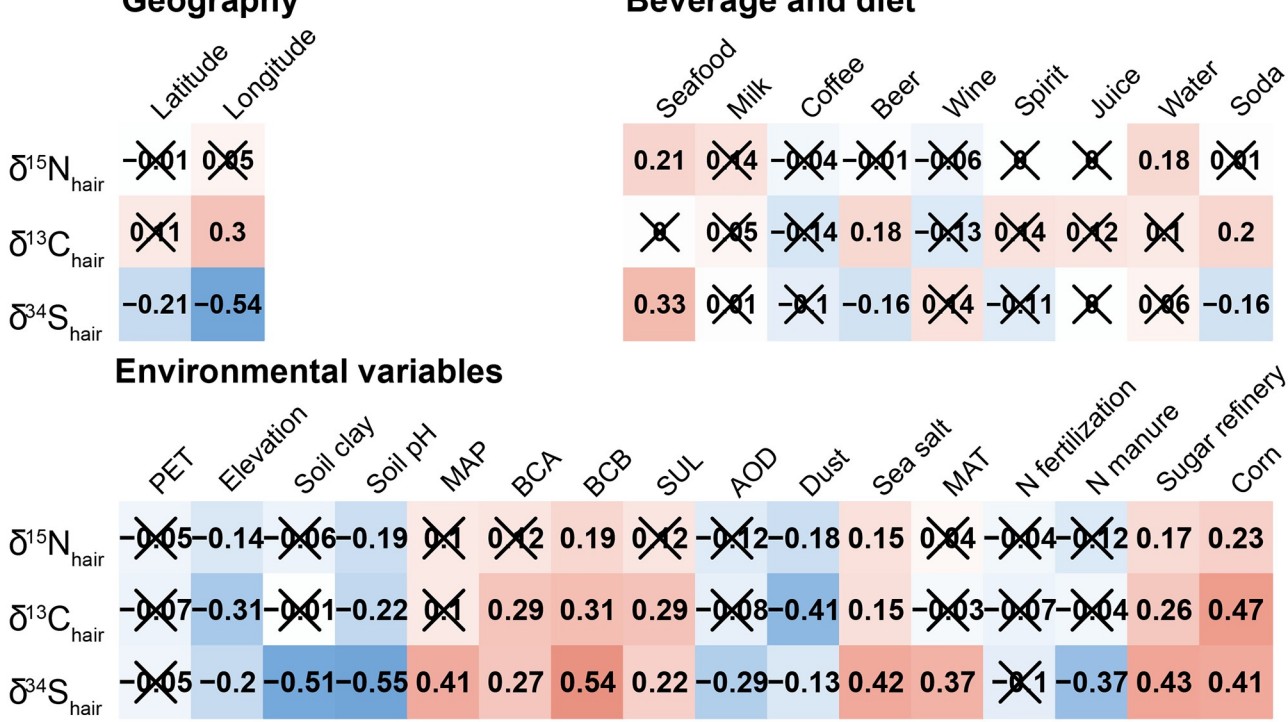

**Fig 3. Pearson correlation coefficient between isotopic data in hair and the continuous dietary choice, geographic and environmental variables.** Red and blue squares indicate significant positive or negative correlation (p-value<0.01) whereas crosses indicate no significant correlation (p-value>0.01).

selects seafood consumption and presence of meat in the diet as predictors but can only explain 5% of the variance (Table 6).

$\delta^{13}C_{hair}$ variations correlate positively with multiple dietary choices and demographic variables including the amount of beer, soda and mineral water consumed, and negatively with the amount of coffee consumed (Fig 3). The $\delta^{13}C_{hair}$ values from males (-18.42 ‰ ± 0.60) were also significantly higher than those of females (-18.61 ‰ ± 0.60; t-test p-value = 2.0e-4). The $\delta^{13}C_{hair}$ values from the younger age group 18–29 years (-18.39 ‰ ± 0.48) were significantly higher than those of the 40–49–18.65 ‰ ± 0.60; t-test p-value = 0.00025) and 60–69 age groups (-18.63 ‰ ± 0.61; t-test p-value = 0.037). $\delta^{13}C_{hair}$ values are significantly higher in participants eating meat than in ovo-lacto vegetarians (Table 4). Our most accurate random forest regression model to predict $\delta^{13}C_{hair}$ variability using dietary choices selects seafood consumption and presence of meat in the diet as predictors but can only explain 18% of the variance (Table 6).

$\delta^{34}S_{hair}$ variations are positively correlated with the amount of seafood consumed (Fig 3). Our most accurate random forest regression model to predict $\delta^{34}S_{hair}$ variations using dietary choices selects seafood consumption as a predictor but can only explain 5% of the variance (Table 6).

**Geographic covariates.** We did not identify any spatial autocorrelation for $\delta^{15}N_{hair}$ values suggesting no spatial trends in $\delta^{15}N_{hair}$ variations (Fig 4A). However, when looking at the data by provinces, we show that Maritimes, Ontario and Quebec have significantly lower $\delta^{15}N_{hair}$ values than other provinces (Table 5 and paired t-test results in S3 Table). Including

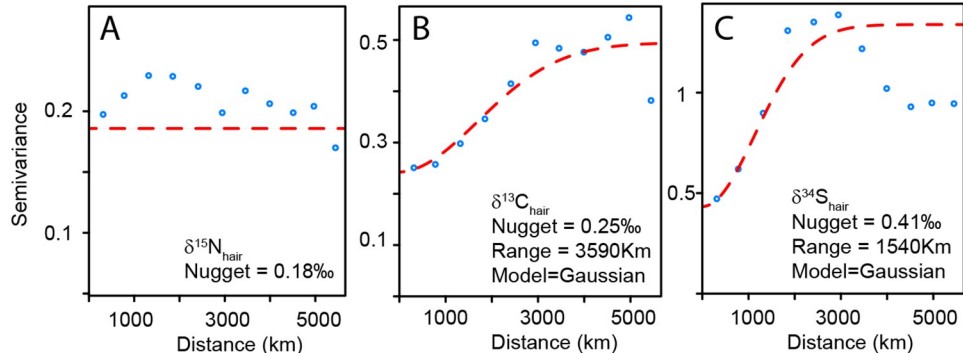

**Fig 4. Semivariograms.** A. $\delta^{15}N_{hair}$ variations; B. $\delta^{13}C_{hair}$ variations; C. $\delta^{34}S_{hair}$ variations. The x-axis distance represents the distance between pairs of observations. The blue points represents the average value of semivariance between point pairs for each 500km distance bin. The red lines represent theoretical semivariograms. Note the nugget value is approximately equal to the analytical precision of 0.2‰, 0.2‰, and 0.4‰ for $\delta^{15}N_{hair}$, $\delta^{13}C_{hair}$, and $\delta^{34}S_{hair}$ values, respectively.

geographic covariate (longitude) in the random forest regression model improve $\delta^{15}N_{hair}$ predictions though the amount of variance explained remains very small (Table 6).

We identified a spatial autocorrelation for $\delta^{13}C_{hair}$ values with a broad spatial range of ~3,500km (Fig 4B). The $\delta^{13}C_{hair}$ values showed higher values in eastern Canada than in western Canada. $\delta^{13}C_{hair}$ values differed significantly between most Canadian provinces (Table 5 and paired t-test results in S4 Table). Including geographic covariates (longitude and latitude) in the random forest regression model significantly improves the $\delta^{13}C_{hair}$ predictions (Table 6).

We identified a strong spatial autocorrelation for $\delta^{34}S_{hair}$ values with a range shorter than that of $\delta^{13}C_{hair}$ values ~1,500km (Fig 4C). The $\delta^{34}S_{hair}$ values showed a decreasing gradient from the coast to more inland locations (Fig 2C). The $\delta^{34}S_{hair}$ values differed significantly between all Canadian provinces (Table 5 and paired t-test results in S4 Table). Including geographic covariates (longitude and latitude) in the regression model strongly improves $\delta^{34}S_{hair}$ predictions (Table 6).

**Dietary choices, demographic and environmental covariates.** $\delta^{15}N_{hair}$ values correlate weakly to sea salt deposition and dust deposition (Fig 3), but including those variables in the regression does not improve the accuracy of $\delta^{15}N_{hair}$ predictions (Table 6).

**Table 5. $\delta^{13}C_{hair}$, $\delta^{15}N_{hair}$ and $\delta^{34}S_{hair}$ values from participants residing in different provinces.**

| Province | n | Average $\delta^{13}C \pm SD$ (‰) | Average $\delta^{15}N \pm SD$ (‰) | Average $\delta^{34}S \pm SD$ (‰) |
|---|---|---|---|---|
| British Columbia (BC) | 129 | -19.0 ± 0.4 | 9.2 ± 0.4 | 1.6 ± 0.7 |
| Alberta (AB) | 83 | -18.8 ± 0.5 | 9.1 ± 0.4 | 1.4 ± 0.7 |
| Saskatchewan (SK) | 42 | -18.7 ± 0.4 | 9.1 ± 0.6 | 0.7 ± 0.7 |
| Manitoba (MB) | 43 | -18.5 ± 0.5 | 9.3 ± 0.4 | 0.3 ± 0.7 |
| Ontario (ON) | 77 | -18.0 ± 0.5 | 9.1 ± 0.5 | 2.0 ± 0.9 |
| Quebec (QC) | 83 | -18.3 ± 0.5 | 9.0 ± 0.4 | 2.2 ± 0.6 |
| New Brunswick (NB) | 30 | -18.3 ± 0.7 | 9.3 ± 0.4 | 2.6 ± 0.7 |
| Nova Scotia (NS) | 58 | -18.3 ± 0.6 | 9.5 ± 0.3 | 2.6 ± 0.8 |
| Prince Edward Island (PE) | 4 | -18.5 ± 0.7 | 9.4 ± 0.6 | 2.7 ± 0.3 |
| Newfoundland Labrador (NL) | 16 | -18.1 ± 0.4 | 9.4 ± 0.4 | 2.7 ± 0.6 |

**Table 6. Summary of multivariate random forest regression modeling for each isotopic system.** Variables are ranked by importance in the regression model. The predictors have either a significant positive (italics) or negative (underline) correlation with isotope data (Pearson correlation; p-value<0.01). $R^2$ = Coefficient of Determination.

| Best model | $\delta^{15}N_{hair}$ | | $\delta^{13}C_{hair}$ | | $\delta^{34}S_{hair}$ | |
|---|---|---|---|---|---|---|
| | Perf. Metrics | Variables Imp. | Perf. Metrics | Variables Imp. | Perf. Metrics | Variables Imp. |
| **Diet** <br> **Demographic data** | $R^2$ <br> 0.05 | <u>Vegetarian</u> <br> *Seafood* | $R^2$ <br> 0.18 | *Soda* <br> <u>Age</u> <br> *Sex* | $R^2$ <br> 0.05 | *Seafood* |
| **Diet** <br> **Demographic data Longitude Latitude** | $R^2$ <br> 0.12 | <u>Vegetarian</u> <br> *Longitude* <br> *Seafood* | $R^2$ <br> 0.26 | *Longitude* <br> <u>Latitude</u> <br> *Soda* <br> *Sex* | $R^2$ <br> 0.53 | *Longitude* <br> *Seafood* <br> <u>Latitude</u> |
| **Diet** <br> **Demographic data Longitude Latitude Environnemental variables** | $R^2$ <br> 0.12 | <u>Vegetarian</u> <br> *Longitude* <br> *Seafood* | $R^2$ <br> 0.32 | *Corn* <br> *Province* <br> *Soda* <br> <u>Sex</u> | $R^2$ <br> 0.62 | <u>Precipitation</u> *Salt aerosol* <br> *Seafood* <br> <u>Soil pH</u> |

$\delta^{13}C_{hair}$ and $\delta^{34}S_{hair}$ values are both strongly correlated to multiple environmental variables (Fig 3) and including environmental covariates in the regression improves the accuracy of predictions (Table 6). However several of those environmental variables are collinear. Using the VSURF algorithm [60], we removed collinear covariates to produce the most accurate $\delta^{13}C_{hair}$ and $\delta^{34}S_{hair}$ predictions with the least variables (Table 6). For $\delta^{13}C_{hair}$ values, the only selected environmental predictor is the distance between volunteer residence and corn agricultural belts (Table 6). For $\delta^{34}S_{hair}$ values, the selected environmental predictors include sea salt aerosol deposition rate, mean annual precipitation, and soil pH ($R^2$, Table 3).

### Isotopic data in resampled participants

In our study, we sampled the volunteers once and their isotopic values represent a snapshot of their last few months of life. We analyzed $\delta^{13}C$, $\delta^{15}N$, and $\delta^{34}S$ values in hair of 25 participants resampled every 6 months over a 4 year period to verify the stability of isotopic values in a given participant and location (S2 Data). Most of the resampled participants show constant $\delta^{15}N_{hair}$, $\delta^{13}C_{hair}$, and $\delta^{34}S_{hair}$ values throughout the sampling period within the range of analytical precision (Table 7). However, a few participants show a higher range of isotopic variability through the resampling period. Three participants for $\delta^{15}N_{hair}$ and $\delta^{13}C_{hair}$, as well as four participants for $\delta^{34}S_{hair}$, have standard deviation through the sampling period higher than twice the analytical precision (Fig 5).

## Discussion

### Isotopic values in Canadian hair compared to other countries

Prior to interpreting the regional isotopic variability within the Canadian dataset, to tease out the influence of dietary choices and geographic controls, we first compared Canadian residents' isotopic signatures to those of other countries (Table 2). The sampled Canadian population shows a limited range of isotopic variability for all isotopic systems (Table 2). In this regard, Canada shows a similar trend to that observed in some other industrialized countries where supermarkets have diminished dietary diversity. This low variability reflects the homogenization of the diet across many industrialized nations as well as the consumption of food products from multiple provenances, which blur the local isotopic baseline. The $\delta^{15}N_{hair}$

**Table 7. Comparison of means and standard deviations of $\delta^{15}N_{hair}$, $\delta^{13}C_{hair}$ and $\delta^{34}S_{hair}$ values for the 25 participants resampled every 6 months across a four-year period (S2 Data).** Numbers highlighted in red correspond to isotopic profiles with a high variance and described in the discussion section. Analytical precision is 0.2‰ for $\delta^{15}N_{hair}$ and $\delta^{13}C_{hair}$ and 0.4‰ $\delta^{34}S_{hair}$.

| | $\delta^{15}N_{hair}$ | | $\delta^{13}C_{hair}$ | | $\delta^{34}S_{hair}$ | |
|---|---|---|---|---|---|---|
| Participant | Mean | Sd | Mean | Sd | mean | Sd |
| 2 | 8.87 | 0.26 | -17.74 | 0.21 | 2.11 | 0.21 |
| 3 | 9.37 | 0.17 | -18.26 | 0.47 | 1.82 | 0.25 |
| 4 | 8.24 | 0.43 | -18.75 | 0.34 | 1.84 | 0.15 |
| 5 | 8.86 | 0.22 | -17.60 | 0.12 | 2.07 | 0.33 |
| 6 | 9.28 | 0.36 | -17.83 | 0.17 | 2.26 | 0.16 |
| 7 | 9.95 | 0.20 | -17.42 | 0.16 | 2.82 | 0.31 |
| 8 | 8.58 | 0.10 | -18.30 | 0.30 | 1.26 | 0.17 |
| 9 | 8.83 | 0.50 | -17.19 | 0.26 | 3.22 | 0.74 |
| 10 | 9.96 | 0.27 | -18.69 | 0.09 | 4.79 | 0.19 |
| 11 | 9.58 | 0.27 | -17.77 | 0.15 | 3.56 | 0.26 |
| 12 | 9.44 | 0.43 | -18.37 | 0.06 | 3.91 | 0.98 |
| 13 | 9.59 | 0.13 | -16.63 | 0.16 | 3.04 | 0.49 |
| 17 | 9.04 | 0.18 | -17.78 | 0.36 | 3.62 | 0.55 |
| 18 | 9.17 | 0.11 | -17.98 | 0.11 | 1.80 | 0.45 |
| 23 | 9.52 | 0.17 | -17.76 | 0.14 | 2.38 | 0.11 |
| 24 | 9.50 | 0.15 | -18.46 | 0.19 | 3.47 | 0.40 |
| 25 | 9.45 | 0.25 | -17.62 | 0.44 | 2.12 | 0.12 |
| 29 | 9.47 | 0.17 | -16.94 | 0.20 | 2.29 | 0.64 |
| 32 | 9.44 | 0.14 | -16.59 | 0.08 | 3.18 | 0.25 |
| 35 | 9.29 | 0.35 | -17.23 | 0.56 | 2.26 | 0.36 |
| 38 | 9.36 | 0.11 | -18.32 | 0.01 | 3.20 | 0.23 |
| 41 | 9.62 | 0.13 | -17.78 | 0.22 | 2.79 | 0.42 |
| 45 | 8.88 | 0.17 | -18.27 | 0.17 | 2.81 | 0.43 |
| 49 | 9.12 | 0.23 | -18.14 | 0.26 | 2.27 | 0.78 |

distribution of Canadians overlaps those found in the USA, and Europe (Table 2). The overlap in these continental distributions suggests in average the same amount of animal protein in diet between these countries. $\delta^{13}C_{hair}$ values in Canada are intermediate between those of Europe and the USA [19,20]. Canadians have lower $\delta^{13}C_{hair}$ values than Americans but higher $\delta^{13}C_{hair}$ values than Europeans (Table 2). Despite similar types of protein sources in the diet of Canadians relative to other industrialized countries, $\delta^{13}C_{hair}$ values in Canadians are distinct, suggesting a different proportion of $C_3$ to $C_4$ plants. Canadian food systems likely have a higher influence of $C_4$ plant by-products than European ones but a lower than American. The $\delta^{34}S_{hair}$ variability is similar to that observed in other industrial countries but the average $\delta^{34}S_{hair}$ values of Canadians is much lower than other countries or regions (Table 2). The lower absolute $\delta^{34}S_{hair}$ values in Canadians suggests that, at least part of the food consumed by Canadian has a distinct $\delta^{34}S$ value relative to other countries. The lower $\delta^{34}S_{hair}$ values found in Canadians are consistent with previous studies in Canadians' hair [51,70], food [51], plants [71] or river water [72]. For example, in a previous study, a self-sustaining human community from Alberta (Hutterite community) had $\delta^{34}S_{hair}$ values close to 0‰ [51] reflecting that of their locally-grown food [51]. In our study, we find similar low $\delta^{34}S_{hair}$ values for Albertan volunteers 1.7±0.7‰ suggesting that locally grown food has a strong influence on Canadian residents' $\delta^{34}S_{hair}$ values. The reasons for the low $\delta^{34}S$ values in food-systems and human in

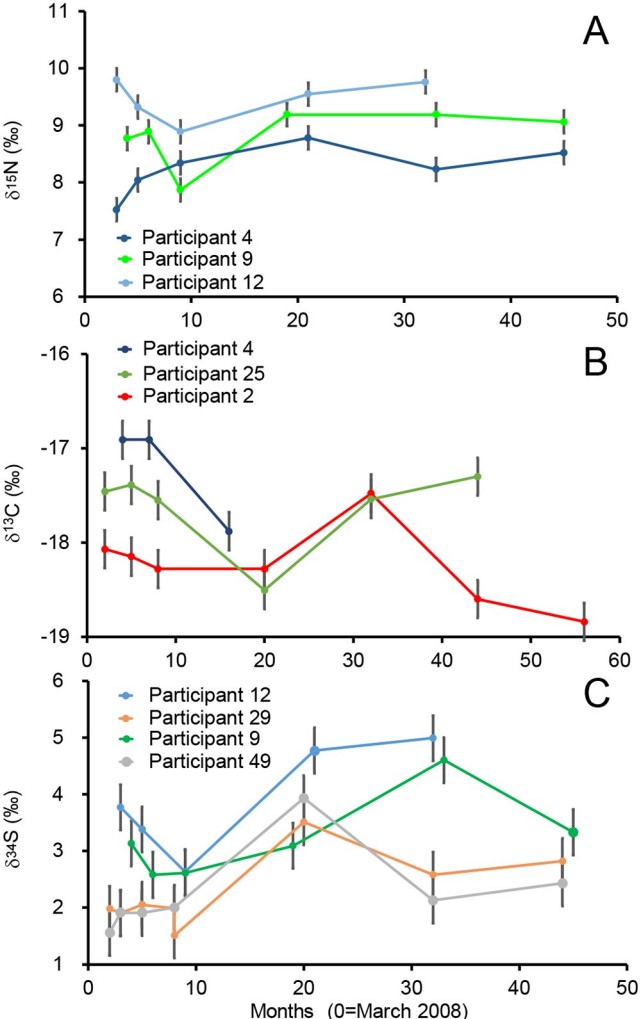

**Fig 5. Isotopic profiles for participants with the most variable isotopic values (Table 7) for A. $\delta^{15}N_{hair}$ values; B. $\delta^{13}C_{hair}$ values and C. $\delta^{34}S_{hair}$ values.**

Canada are likely related to specific geological, climatological and/or anthropogenic controls [73]. Canadian soils receive low amount of isotopically heavy marine sulphates but large amount of isotopically light anthropogenic S from the eastern USA [71]. Canadian farmers use fertilizers produced with ammonium sulphates and ammonium nitrates, which are manufactured using isotopically low crude oils and ore sulfides [54]. Microbially or plant mediated isotopic fractionation amplified by climate conditions could also explain the low $\delta^{34}S$ values in Canadian food systems [52]. In any case, we argue that the low $\delta^{34}S$ values found in Canadian residents' hair reflect the unique isotopic signature of Canadian food products transmitted to human tissues, because more than 70% of the food consumed by Canadian is produced within the country [55,56].

## Dietary choices and demographic controls of isotopic variability

We observed some significant correlations between isotopic variability and several dietary choices. However, these correlations remain weak and are of limited use when trying to predict the isotopic variability at the population scale (Table 6). We underline that this low accuracy is

typical of population studies using self-reported food frequency questionnaire (FFQ). These FFQ are not always valid to represent dietary inputs in volunteers because of the inability of volunteer to fully and accurately recall their intakes [74] and the possibility that individual physiology in metabolizing of food may affect the isotopic values of hair [3]. Self-reported FFQ are also difficult to compare between participants because of intentional misreporting or biased reporting about certain food consumption due to personal characteristics or living conditions [74]. Even in studies using detailed FFQ, the relationship between dietary habits and high-quality dietary biomarkers are often low ($R^2$<0.1) [3]. In our study, the FFQ lacks some important details which have all been shown to influence isotopic variability in human tissues, such as the type of fish or meat consumed, the source of food (e.g., organic vs. conventional), or the amount of legumes consumed. [e.g., 76]. Hence, our approach likely underestimates the role of dietary choices in controlling isotopic variability. Recognizing these limitations and biases, we use our questionnaire as the basis to assess how the reported dietary choices and demographic information relate to isotopic variability.

$\delta^{15}N_{hair}$ values vary with the rate and type of animal-protein consumed, with significantly higher values for Canadians who frequently eat seafood (Fig 3) and significantly lower values in ovo-lacto vegetarians relative to the rest of the population (Table 4). This finding reinforces previous studies that showed the importance of seafood consumption on $\delta^{15}N_{hair}$ variability at the individual [32] and at the population scale [see review in 10]. However, when including seafood and meat consumption as predictors in a multivariate regression model, those variables explain only a negligible portion of the $\delta^{15}N_{hair}$ variance in the Canadian population (Table 6). As mentioned above, better explaining $\delta^{15}N_{hair}$ variability would require more detailed FFQ with information about meat consumption type and rate, the type of fish and seafood consumed, the amount of legumes in the diet, the source of additional protein intakes, or the rate of consumption of organic food.

Canadian ovo-lacto vegetarians have significantly lower $\delta^{13}C_{hair}$ values than meat-eaters (Table 4). Most of the dietary carbon in Canadians likely comes from animal proteins and cereal consumption. As part of the Canadian livestock is corn fed, Canadian meat-eaters have a higher proportion of $C_4$ plants in their diet relative to non-meat eaters [75]. Canadians who consume more soda and beer have higher $\delta^{13}C_{hair}$ values (Fig 3). Soda and sweet consumption can make up a significant part of dietary carbon sources [76]. As most sugar and sweeteners in Canada are based on $C_4$ plants, increased sugar consumption contributes to increased $\delta^{13}C_{hair}$ values [1]. Some demographic variables, specifically sex and age of Canadians, also show significant relationships with $\delta^{13}C_{hair}$ values (Table 3). These variables likely reflect the preferential consumption of meat and sugar by young Canadian males relative to the rest of the population [77]. However, taken together, these dietary choices and demographic variables explain less than 20% of the $\delta^{13}C_{hair}$ variance in the population. While this is higher than for $\delta^{15}N_{hair}$ variations, this low predictive power likely underestimates the role of dietary choices in controlling $\delta^{13}C_{hair}$ variability due to the limitations of the FFQ.

As expected, $\delta^{34}S_{hair}$ values increase with seafood consumption rate (Fig 3). The $\delta^{34}S$ values in seafood products are much higher than terrestrial food as it reflects that of seawater (~20‰). However, even in coastal localities and for participants with high seafood consumption, the $\delta^{34}S_{hair}$ values of Canadians remain quite lower than that of the ocean (Fig 2C). This is a bit surprising because in other regions of the world, $\delta^{34}S$ of participants consuming a high amount of seafood tend to be much higher [19]. In hair, most of the sulphur comes from cysteine, a non-essential amino acid that can be biosynthesised from methionine and serine or assimilated directly from diet [78]. Many food items other than seafood can contribute to cysteine assimilation in the human body, including meat, eggs, dairy products, or cereals [79]. This relatively low sulphur contribution of seafood consumption in the modern human diet

probably explains why the $\delta^{34}S_{hair}$ of Canadian living on the coast is low. As mentioned above (Table 2), Canadians probably consume a large amount of glocal food with low $\delta^{34}S$ values buffering the $\delta^{34}S_{hair}$ values of heavy seafood consumers. We conclude that while several dietary choices are significantly related to Canadian' isotopic variability, particularly for $\delta^{13}C_{hair}$ values, more detailed FFQ are required to fully capture this variability at the population scale.

## Geographic and environmental controls of isotopic variability

We show that in the Canadian population, geographic and environmental variables have an influence on all isotopic systems. The influence of these geographic variables is weakest for $\delta^{15}N_{hair}$ variations. The only observed geographic trend in $\delta^{15}N_{hair}$ variations is that Canadians from the Maritimes have significantly higher $\delta^{15}N_{hair}$ values than those from other provinces (Table 5). This trend likely reflects the higher production and consumption of seafood in regions close to the coast. The remaining distribution of $\delta^{15}N_{hair}$ values is spatially random as evidenced by the absence of spatial autocorrelation (Fig 4A). The lack of spatial trend is surprising because food systems in Canada should show a broad range of $\delta^{15}N$ values reflecting the large range of climatic conditions, soil type and agricultural practises across the country [34,36,37]. In particular, different agricultural centers in Canada use different fertilizers (e.g., manure vs. manufactured fertilizers) and different amount of legumes in livestock feed (e.g., soybean and corn vs. barley) (http://open.canada.ca/data). If Canadians ate dominantly glocal food, this isotopic variability should be propagated into Canadian consumers. Interestingly, $\delta^{15}N_{hair}$ values correlate best with the distance to major corn production zones (Southeast Ontario and Saint Lawrence River Valley) (Fig 3). Ontario and Quebec have slightly lower $\delta^{15}N_{hair}$ than other provinces (Table 5). Both provinces produce the great majority of soybean crops in Canada (http://open.canada.ca/data). The low $\delta^{15}N$ of soybean (i.e., a legume) is potentially transmitted in livestock raised in these provinces, and ultimately in human food products. Our data shows that $\delta^{15}N_{hair}$ values have a very limited relationship with environmental conditions across Canada (Table 6). Most of the $\delta^{15}N_{hair}$ variance of this Canadian population remains unexplained. As suggested in a previous study [70], $\delta^{15}N_{hair}$ values have almost no relationship with geography and a very low potential at discriminating the geographic origin of participants.

The geographic location of residence influences the $\delta^{13}C_{hair}$ signatures of hair donors, suggesting a link between humans and local agri-food systems in Canada (Table 6). The spatial autocorrelation for $\delta^{13}C_{hair}$ data is regional in scale, as evidenced by the large range of the semi-variogram (Fig 4B). Canadians from eastern provinces have significantly higher $\delta^{13}C_{hair}$ values than those from western provinces (Figs 1B and 6). $\delta^{13}C_{hair}$ values correlate with the distance to major corn production zones (Southeast Ontario and Saint Lawrence River Valley) (Table 6). Eastern Canada produces the great majority of Canadian corn (a $C_4$ plant with high $\delta^{13}C$ values). This corn is the primary source of feed for livestock in eastern Canada, particularly in Ontario (https://www.anacan.org/). As for soybean, this corn and its by-products (e.g., cornstarch, syrup, sweeteners, or oil) are also used throughout the food processing industry of eastern Canada. Food products such as corn-fed livestock (e.g., meat, milk), and corn-rich processed food (e.g., sodas) have a high $\delta^{13}C$ value [27]. Conversely, western Canada grows little corn but grows most of the wheat and barley of Canada (both $C_3$ plants with low $\delta^{13}C$ values). This wheat and barley are the dominant source of feed for livestock(https://www.anacan.org/). This regional $\delta^{13}C_{hair}$ pattern is not extremely surprising considering that the supply-managed commodities (eggs, meat, milk) are usually produced within the province and contribute to a large portion of the dietary carbon [56]. Ultimately, the isotopic signatures of human tissues across Canada reflect the type of major crops grown in food systems within their province

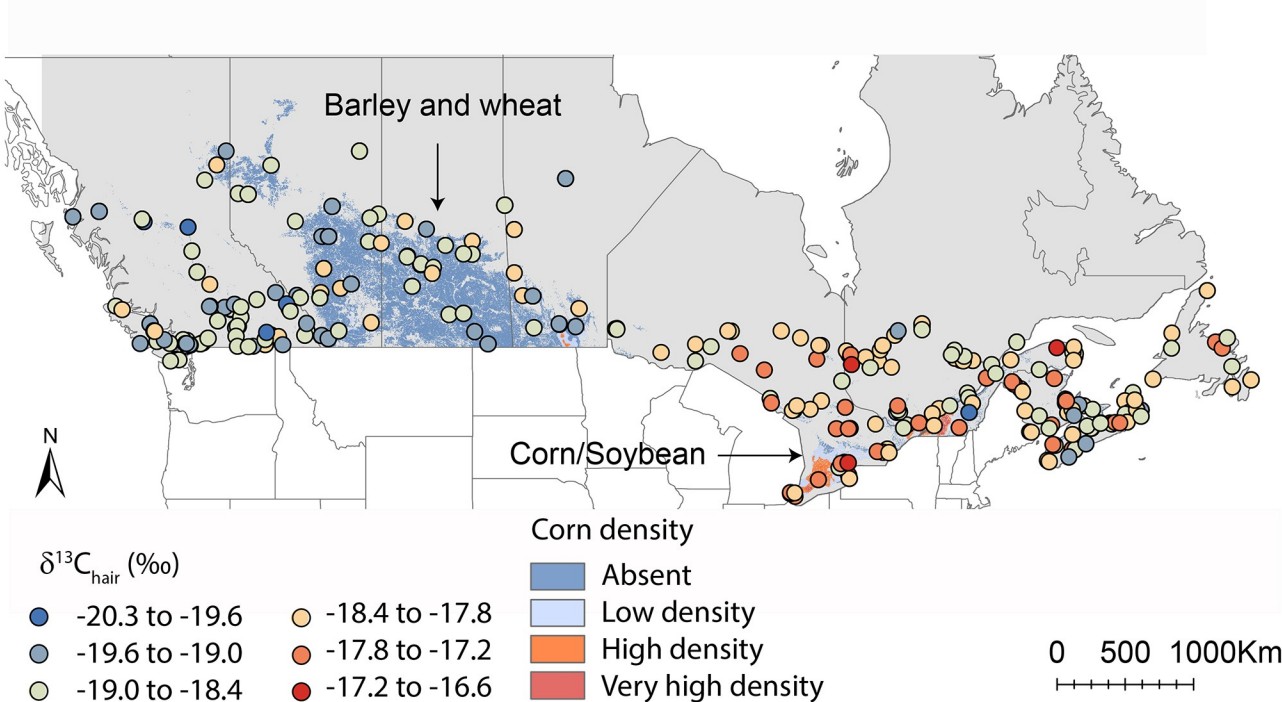

**Fig 6. Distribution of carbon isotope variations in hair of donors and density of corn production across Canada.** Color scale represents the spatial density of corn crops on agricultural land relative to other $C_3$ crops for the year 2011 (http://open.canada.ca/data). This map contains information licensed under the Open Government Licence–Canada. Administrative boundaries are from http://www.naturalearthdata.com/. This map was generated in Rx64 3 4.2 (https://www.r-project.org/).

(Fig 6). This observation is consistent with the high consumption of food produced using local agricultural goods in Canada [56]. Our multivariate regression could explain 32% of the total $\delta^{13}C_{hair}$ variations in the Canadian population. While this is better than for $\delta^{15}N_{hair}$ values, the majority of the $\delta^{13}C_{hair}$ variance remains unexplained. As mentioned earlier, the main reason for this lack of predictive power is likely the limitations of our FFQ which does not report for some key dietary choices.

$\delta^{34}S_{hair}$ values vary at higher spatial resolution than $\delta^{13}C_{hair}$ values with a spatial autocorrelation range of 1,500km (Fig 4C). The $\delta^{34}S_{hair}$ values differ significantly between each Canadian province (t-test; p-value<0.01). $\delta^{34}S_{hair}$ values decrease progressively from coastal to more inland regions (Fig 2C). As for other isotopic systems, the reported dietary choices explain only a very limited amount of $\delta^{34}S_{hair}$ variance (Table 6; Fig 7A). However, when integrating geographic and environmental covariates, our random forest regression can explain the majority of the $\delta^{34}S_{hair}$ variance. A model including latitude and longitude along with seafood consumption rate explains 53% of the variance (Table 6; Fig 7B). The model explains 62% of the variance when including sea salt deposition, soil pH, seafood consumption, and precipitation as predictors (Table 6; Fig 7D). When only including environmental variables and removing all dietary choice variables, we found that the model still explained 55% of the $\delta^{34}S_{hair}$ variance (Table 6; Fig 7C). In contrast to other isotopic systems, $\delta^{34}S_{hair}$ variability in Canadians is strongly correlated with geographic and environmental predictors. Interestingly, the trend observed in the $\delta^{34}S_{hair}$ values across Canada correlates with the expected spatial $\delta^{34}S$ variability in local food systems. $\delta^{34}S$ in Canadian crops should reflect the mixture of several isotopically-distinct S sources: 1) isotopically light sulphates from the soil solution [73], 2)

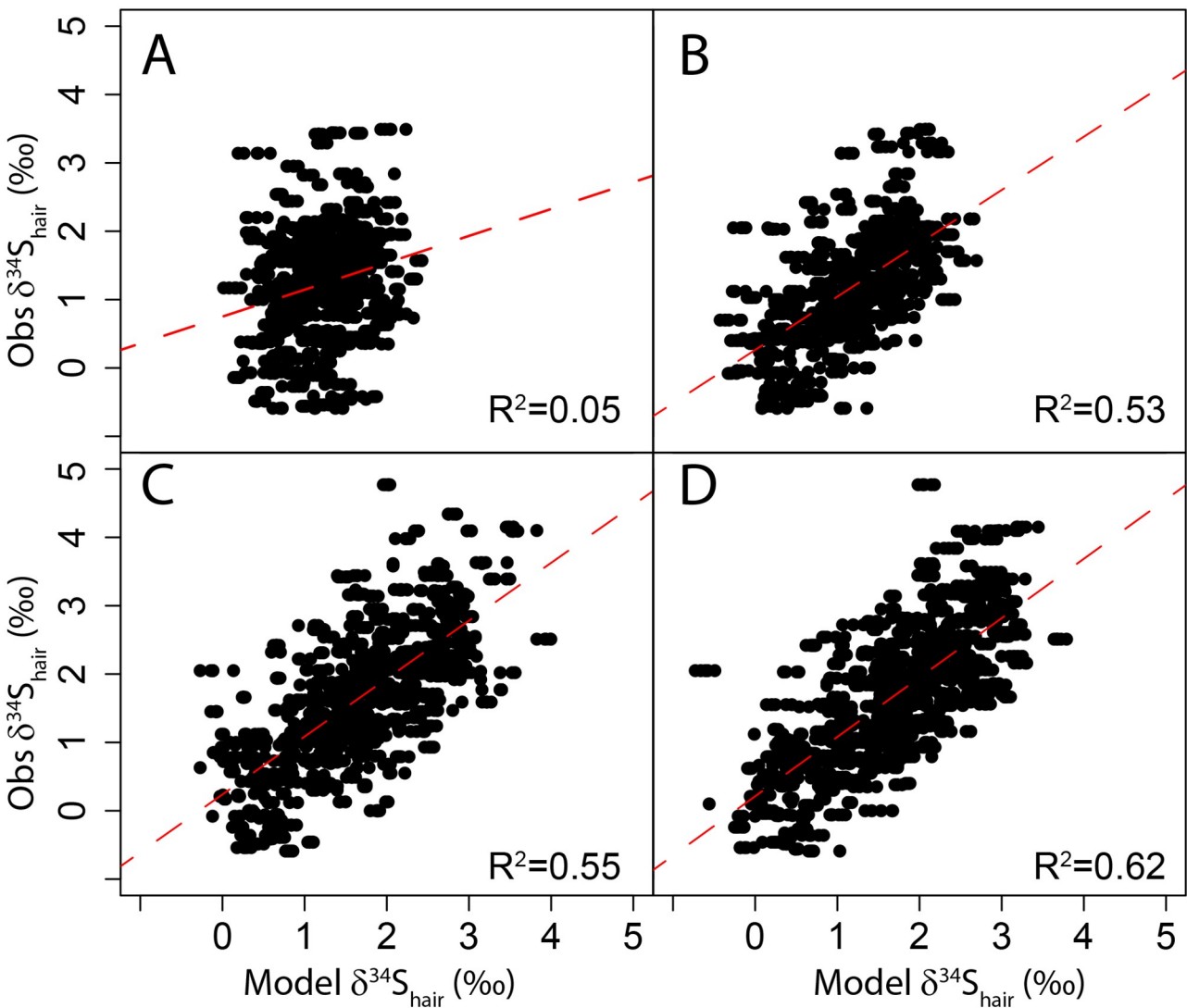

**Fig 7. Observed $\delta^{34}S_{hair}$ against modeled $\delta^{34}S_{hair}$.** A. Dietary and demographic data regression model including seafood, wine, and water consumption; B. Dietary, demographic and geographic data regression model including longitude, seafood, wine, and water consumption; C. Environmental variables data regression model including sea salt aerosol deposition and soil pH; D. Dietary, demographic, geographic and environmental variable regression model including sea salt aerosol, soil pH, precipitation and seafood consumption. The red dashed line is the best fit linear model.

isotopically heavy marine aerosols [71,80] and 3) isotopically light sulphates from atmospheric pollutants and fertilizers [71]. As these isotopically distinct sources are mixed in soils, isotopic fractionation by microbial processes and plant metabolism might further modify their isotopic composition [52]. In several locations of the world, $\delta^{34}S$ in soils, plants or even livestocks exhibit a distinct spatial pattern of decreasing values towards more inland locations [49,71,80,81]. In Canadian coastal provinces, food systems should have high $\delta^{34}S$ values because acid and saline soils are dominated by marine sulphates [80]. As Canadian food systems become more distant from the coast, bedrock S or anthropogenic sources dominate decreasing $\delta^{34}S$ values [71]. In alkaline soils of the Prairies, food systems should uptake most of their S from geological sources or from anthropogenic fertilizers with low $\delta^{34}S$ values [73]. The $\delta^{34}S_{hair}$ variability in Canadians closely follows this expected isotopic pattern in food

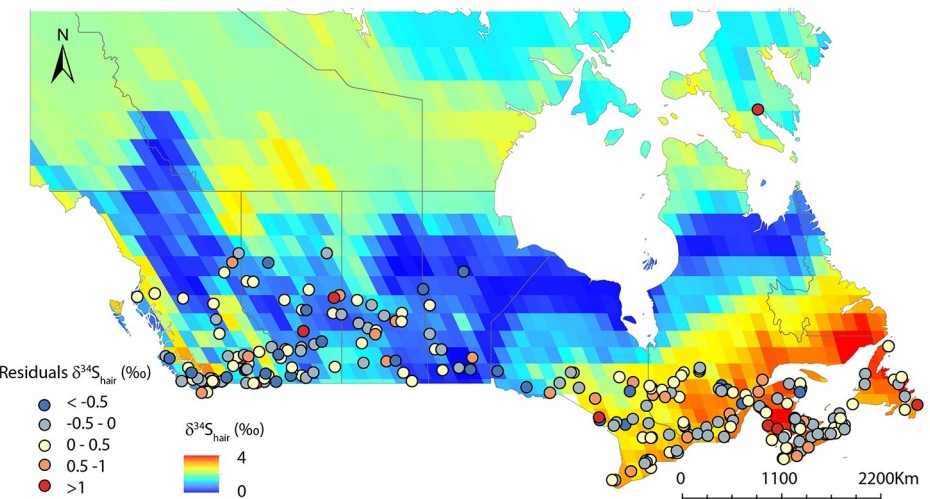

**Fig 8. Predicted spatial $\delta^{34}S_{hair}$ variability using rainfall, salt aerosol, and soil pH as predictors (resampled at 10,000km$^2$).** Colored points represent the associated residuals (modeled $\delta^{34}S_{hair}$ values–observed $\delta^{34}S_{hair}$ values). Administrative boundaries are from http://www.naturalearthdata.com/. This map was generated in Rx64 3 4.2 (https://www.r-project.org/).

systems (Fig 8). We hypothesize that this spatial isotopic trend in the hair of Canadian residents is consistent with a high percentage of intra-provincial food consumption in Canadian markets [55]. Canadian customers likely obtain a large part of their cysteine S from locally sourced high-protein food items (e.g., meat, yogurt, cheese, eggs, farmed fish). As observed in other countries [49], these animal products inherit their $\delta^{34}S$ from that of regionally-grown crops at the base of food systems. Even though humans have complex dietary habits, this regional $\delta^{34}S$ is transmitted to human hair due to the dominance of local to regional food in retailing stores [56].

This strong geographic/environmental dependence of $\delta^{34}S_{hair}$ values across Canada reinforces the idea that $\delta^{34}S$ values could be very useful in geological applications in archeology and forensic sciences [44]. To illustrate these applications, we calculate the residuals between observed and predicted $\delta^{34}S_{hair}$ variability (Fig 8). We demonstrate that high $\delta^{34}S_{hair}$ residuals are associated with participants with very high consumption of seafood products or with participants who traveled to a distant location in the recent past. For example, several of the positive residuals in Fig 8 are from participants who traveled to Europe, Florida, or the Caribbean within the last 3 months before collection (S2 Data). Despite our careful sampling procedure (Materials and Methods), the hair from these participants likely contains some S from non-Canadian food sources with higher $\delta^{34}S$ values as hair does not grow at the same rate and time [57] (Table 7). Identifying participants or local populations deviating from the expected $\delta^{34}S_{hair}$ trends could become useful in a range of applications from tracing the proportion of local food consumed to reconstructing the travel history of participants. On a global scale, $\delta^{34}S_{hair}$ values might also become useful to track the amount of imported food consumed. In countries with little agricultural production (e.g., Middle East), $\delta^{34}S_{hair}$ values should be inherited from imported food. Other countries, such as Costa Rica, Bolivia, Peru, and Japan also show low $\delta^{34}S_{hair}$ values [70]. The food $\delta^{34}S$ baseline in these countries is likely influenced by the high contribution of isotopically low geological S from volcaniclastic sediments and ashes [82]. While traditionally $\delta^{34}S_{hair}$ values had been primarily used as a proxy for seafood consumption [19,20], the results from this study indicate a strong potential of $\delta^{34}S_{hair}$ for geolocation and food traceability studies.

## Isotope variability in hair through time

Despite the rapid growth rate of human hair, we show that for most of the resampled volunteers, there is little isotopic variability over a period of 4 years (Table 7). This stability of isotopic signals in Canadian human tissues is encouraging and suggest that the spatial patterns observed in our study are stable. The lack of temporal isotopic variability in volunteers probably reflect the stability of dietary habits, physiology and isotopic baselines of the food consumed in the hair donors. The few participants that showed a higher isotopic variability reported either a recent trip or a dietary change over the period of sampling (S2 Data). Out of the 3 participants with highly variable $\delta^{15}N_{hair}$ values (Fig 5A), participant 4 stopped eating dairy products for a prolonged period between month 25 and 35, while the other two (participants 9 and 12) were frequent but variable seafood consumers. Participants 9 and 12 also showed variable $\delta^{34}S_{hair}$ values, reinforcing the link between seafood consumption and $\delta^{15}N_{hair}/\delta^{34}S_{hair}$ variability (Fig 5C). Two other participants (29 and 49) with variable $\delta^{34}S_{hair}$ values lived together throughout the resampling experiment and traveled together within Canada (Fig 5C). Interestingly, these two participants show identical $\delta^{34}S_{hair}$ variability, further reinforcing the idea that $\delta^{34}S_{hair}$ values may be dominantly controlled by the geographic origin of the diet and not by physiology or diet choices. For the three participants with more variable $\delta^{13}C_{hair}$ values, two of those (participants 2, 25) reported a recent trip within the sampling period. Participant 2 shows a high $\delta^{13}C_{hair}$ value within its profile, likely denoting a 2 weeks long trip to Florida; whereas participant 25 shows a low $\delta^{13}C_{hair}$ value within its profile, likely corresponding to a multi-weeks long trip to Alberta. The last participant with more variable $\delta^{13}C_{hair}$ values did not report its traveling history, which complicates interpretation.

## Conclusions

As expected, dietary choices can influence $\delta^{15}N_{hair}$, $\delta^{13}C_{hair}$, and $\delta^{34}S_{hair}$ variability at the population scale. However, more accurate and detailed FFQ's are required to capture the full influence of dietary choices on each isotopic system. More interestingly, we found that for $\delta^{34}S_{hair}$ and to a lesser degree $\delta^{13}C_{hair}$ values, a large portion of the isotopic variability is explained by the location of residence of volunteers. In particular, $\delta^{34}S_{hair}$ values display predictable patterns across Canada that follows that of local food systems. We hypothesize that these patterns reflect the specific isotopic signatures of regional food systems across Canada transmitted to human tissues through the consumption of glocal food. Our study underlines the importance of local isotopic food baselines in controlling some of the isotopic variability across a population. Our work also paves the way for promising applications of S isotopes in food and forensic science.

## Supporting information

**S1 Data. Excel data table with C, N and S isotopes data in hair of 590 participants.**
(XLS)

**S2 Data. Excel data table with C, N and S isotopes data in hair of 25 resampled participants.**
(XLSX)

**S1 Script. R code detailing the statistical analysis conducted in this study.**
(R)

**S1 Table. Demographics and dietary questions answered by the volunteers to the collection scientist.** Note: samples were collected across several years and some dietary questions were

only added post year 1 of the collection efforts.
(DOCX)

**S2 Table. p-values from Shapiro tests assessing the normality of $\delta^{13}C_{hair}$, $\delta^{15}N_{hair}$ and $\delta^{34}S_{hair}$ distribution.** p-value>0.05 indicates the distribution is not significantly different from normality.
(DOCX)

**S3 Table. p-values from t-tests comparing hair $\delta^{15}N_{hair}$ values from different provinces.** p-values less than 0.05 are highlighted in grey. Values in italics represent provinces with unequal variance (Levene's test).
(DOCX)

**S4 Table. p-values from t-tests comparing $\delta^{13}C_{hair}$ values from different provinces.** p-values less than 0.05 are highlighted in grey. Values in italics represent provinces with unequal variance (Levene's test).
(DOCX)

**S5 Table. p-values from t-tests comparing hair $\delta^{34}S_{hair}$ values from different provinces.** p-values less than 0.05 are highlighted in grey. Values in italics represent provinces with unequal variance (Levene's test).
(DOCX)

## Acknowledgments

We thank J. Ehleringer for helpful discussions. We thank Dorothée Drucker and two anonymous reviewers for their help in improving this manuscript.

## Author Contributions

**Conceptualization:** Clement P. Bataille, Michelle M. G. Chartrand.

**Data curation:** Clement P. Bataille, Michelle M. G. Chartrand, Gilles St-Jean.

**Formal analysis:** Clement P. Bataille, Francis Raposo.

**Funding acquisition:** Clement P. Bataille, Michelle M. G. Chartrand, Gilles St-Jean.

**Investigation:** Clement P. Bataille, Gilles St-Jean.

**Methodology:** Clement P. Bataille, Michelle M. G. Chartrand.

**Project administration:** Clement P. Bataille, Michelle M. G. Chartrand, Gilles St-Jean.

**Resources:** Clement P. Bataille, Michelle M. G. Chartrand.

**Software:** Clement P. Bataille.

**Supervision:** Gilles St-Jean.

**Validation:** Clement P. Bataille, Michelle M. G. Chartrand, Francis Raposo.

**Visualization:** Clement P. Bataille, Michelle M. G. Chartrand, Francis Raposo, Gilles St-Jean.

**Writing – original draft:** Clement P. Bataille.

**Writing – review & editing:** Clement P. Bataille, Michelle M. G. Chartrand.

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
