## [Decision Letter · Decision Letter 0]

27 May 2020

PONE-D-20-08053

Disentangling Dietary and Non-Dietary Controls of Hair Isotopic Variability in Human Populations: A case-study in Canada

PLOS ONE

Dear Dr. Bataille,

Thank you for submitting your manuscript to PLOS ONE. After careful consideration, we feel that it has merit but does not fully meet PLOS ONE’s publication criteria as it currently stands. Therefore, we invite you to submit a revised version of the manuscript that addresses the points raised during the review process.

We look forward to receiving your revised manuscript.

Kind regards,

Dorothée Drucker

Academic Editor

PLOS ONE

Additional Editor Comments:

Both reviewers underlined the importance of the data set presented in this paper. However, several points need to be thoroughly examined. The reliability and pertinence of the questionnaire should be further discussed and evaluated. Some improvement about information category may be suggested in regard to the results and difficulties in the interpretation. The reviewer 1 raised an important point with the normalization of the isotopic results that has to be addressed by the authors, especially for the evaluation of possible inter-comparison with other studies on human hair. The authors are also invited to consider additional bibliographic resources about human hair growth and isotopic recording. Discussion on the dietary vs non-dietary control on the 13C and 15N abundances in hair needs to be improved. The hypothesis relating to physiological stress seems to be unlikely over a large period of time and geographical range. Elements are missing: data in Table 2 (see reviewer 2), information in the R script, (see reviewer 1), Pearson correlation coefficients in supplementary tables.

Journal Requirements:

Reviewers' comments:

Reviewer's Responses to Questions

**Comments to the Author**

1. Is the manuscript technically sound, and do the data support the conclusions?

Reviewer #1: Yes

Reviewer #2: Partly

2. Has the statistical analysis been performed appropriately and rigorously? 

Reviewer #1: N/A

Reviewer #2: I Don't Know

3. Have the authors made all data underlying the findings in their manuscript fully available?

Reviewer #1: No

Reviewer #2: Yes

4. Is the manuscript presented in an intelligible fashion and written in standard English?

Reviewer #1: Yes

Reviewer #2: Yes

5. Review Comments to the Author

Reviewer #1: This paper delivers data of δ13C, δ15N and δ34S values in hair samples from Canadian volunteers from all over the country. It contains isotopic information about self-reported composition of diet, different climates, soils and food components over Canada. At the periods of hair sampling the volunteers answered different questions, mainly about their dietary habits. Based on individual dietary and additional non-dietary information, the authors present all data. The data were statistically evaluated by machine-learning regression systems.

The author present important data, which were evaluated very well, and they present well-researched information. Furthermore, the authors developed excellent spatial distributions of hair isotope values.

After major revisions of the manuscript this paper should be published in PlosOne.

INTRODUCTION

Page 2, line 42:

In addition, δ15N values in diet and consequently in human tissues are influenced by the intake of legumes and the kind of fertilizer (organic vs. synthetic) used for food production, these factors should also be mentioned at that point.

In Principle: How is the situation in Canada regarding soya? What’s about its consumption by the Canadians - directly from Soya products or via meat from animals, which were fed by soya?

Page 2, line 46:

The statement could be that δ34S values may also be influenced by trophic level. As I understood from the mentioned references, shifts of δ34S within the food web often are within the analytical uncertainties and the results are not always comprehensible (see also Tanz, N., & Schmidt, H. L. (2010). δ34S-value measurements in food origin assignments and sulfur isotope fractionations in plants and animals. Journal of agricultural and food chemistry, 58(5), 3139-3146.)

Page 3, line 62: The paper from Arneson and McAvoy 2005 deals with isotope fractionation from diet to mouse tissues. These results are not directly comparable with the dietary situation and metabolism of modern human adults. I my opinion it is not necessary to mention isotope fractionation from diet to humans at this point, therefore you may delete the sentence in the text.

Page 3, line 64: There are some more references dealing with time-resolved incorporation of several isotopes into hair, which are worth mentioning:

• Huelsemann, F., Flenker, U., Koehler, K., & Schaenzer, W. (2009). Effect of a controlled dietary change on carbon and nitrogen stable isotope ratios of human hair. Rapid Communications in Mass Spectrometry: An International Journal Devoted to the Rapid Dissemination of Up‐to‐the‐Minute Research in Mass Spectrometry, 23(16), 2448-2454.

• Lehn, C., Lihl, C., & Roßmann, A. (2015). Change of geographical location from Germany (Bavaria) to USA (Arizona) and its effect on H–C–N–S stable isotopes in human hair. Isotopes in environmental and health studies, 51(1), 68-79.

• Lehn, C., Kalbhenn, E. M., Rossmann, A., & Graw, M. (2019). Revealing details of stays abroad by sequential stable isotope analyses along human hair strands. International journal of legal medicine, 133(3), 935-947.

Page 3, line 71-73: Both references “33” and “35” are identical.

Page 3, line 73 and page 4, line 75: Reference “35” is wrong at these points.

Page 5, line 101: Please check if citation “48” is right at that point.

The following sentence: As mentioned above, for clarification all the “dietary” factors should be presented in a table.

Page 5 line 99ff: You may shorten the text by deletion of the whole part “Regions like Brazil ... .... not unique”.

Line 105ff: As mentioned above, if you want to utilize the terms “dietary” and “non-dietary”, there is a need for an exact definition. For example, “dietary”: depending on the composition of diet, and “non-dietary”: depending on agricultural production methods (and environmental/ climatic conditions?)

MATERIALS AND METHODS

Page 8, line 173: As mentioned below, you may delete the sentence “C, N and S ...... isotopic fractionation [14].”

Page 8, line 182ff: It seems that for calibration of hair δ13C, δ15N, and δ34S values only inorganic standards were used. However, for analyzing hair samples keratin standards should be used for calibration. The authors mentioned they used in-house hair standards, but no isotope values are given. Why have you not taken USGS42 and USGS43 standards for calibration or at least for comparison with the results of your in house standards? In general, isotope values of hair samples analyzed by different groups or laboratories should be comparable, and this can only be achieved by the use of international isotope standards being of the same material (hair, at least keratin).

Page 9, Supplementary data:

In my download of the Plosone_SI.docx file, the Pearson correlation coefficients of Tables S3, S10 and S15 are not pictured.

Unfortunately I am not familiar with “R”, because I am using another statistical program (SPSS). I asked a colleague to open the R script. She found that “supl data called Data S1 does not include a sheet required for the script to run (‘Dietary’).”

RESULTS

Table S6, S8 and S11 should be presented in the manuscript. Provinces in Table S11 should be named.

Chapter “Isotopic data in Canadian hair compared to other countries”

Page 15, line 292: The reference of Lehn et al. 2015 should be considered as well. It contains many δ13C, δ15N and δ34S data of worldwide collected human hair samples

• Lehn, C., Rossmann, A., & Graw, M. (2015). Provenancing of unidentified corpses by stable isotope techniques–presentation of case studies. Science & Justice, 55(1), 72-88).

Page 16, line 323: incorrect spelling of δ34S

For data evaluation the authors differentiate between “dietary” and “non-dietary” factors, but I could not find an exact definition thereof. It seems that “dietary” factors are based on individual information the volunteers replied to the questionnaires. However, the answers of the volunteers contain information about the composition of their diet, but also e.g. about smoking habits and possible hair dying. The authors should clarify the content or meaning of “dietary” factors used for data evaluations. This may avoid misunderstandings. For example, the proportion of C3 or C4 plants in diet could be considered as a “dietary” factor, but in the paper the proportion of C3 or C4 plants belongs to the “non-dietary” factors.

DISCUSSION

Page 23, line 401: “The overlapping of δ15N hair distribution between Canada, Europe and USA suggests a similar diet...” I do not agree to this statement, because “diet” or composition of diet in these regions is different. My suggestion: ... suggests in average the same amount of animal protein in diet and similar agricultural production methods for terrestrial food products between these countries.

Page 23, line 416: Change the sentence as follows: “The reasons for the low δ34S values in food-systems and human in Canada are likely related to...”

Furthermore, the kind of fertilizer may lead to low δ34S values. Which was the mostly applied fertilizer in Canada, and where did it come from?

Page 23, line 419: My suggestion: “... isotopically light S of anthropogenic sources (do you mean coal combustion?) from the eastern USA ...” Please check if this statement actually exists in the mentioned reference [45].

Page 24, line 433: It is very important to mention that self-reported diet is difficult to compare between the individuals. The personal specifications are not objective, but mostly relative. For example, I have the experience that people from the coast often underestimate their consumption of sea products, whereas people from regions in the inland, where it is rather unusual to eat plenty of sea products, overestimate its consumption.

In The FFQ some more personal details of diet are lacking: preference of organic or conventional food, amount of legumes in diet, preference of sea fish or freshwater fish.

Page 25, line 40: Please add some non-dietary factors that may explain δ15N hair variability, e.g. kind of fertilizer, agricultural production methods, amount of legumes in diet, intake of additional protein, health status.

Page 25, line 449: What is about the intake of animal protein from dairy products in non-meat eaters?

Page 27, line 493: You may delete this sentence “Most .... unexplained.”

Just a brief comment: In accordance with your results relating to δ15N hair values, the results in Lehn et al. 2015 (Table 2) indicate that δ15N values in hair samples have the lowest potential (among C-N-S-H stable isotope values) for geographical discrimination into the different groups of origin. The statistical evaluations of the C-N-S-H stable isotopes in human hair samples were performed by canonical discriminant analyses.

Page 27, line 497: Please exchange “disease” by “certain diseases”.

Page 27, line 498ff: The following statement “the broad range of metabolism and health conditions between the individuals ... drive most of the δ15N hair variations” should be modified. Changes in metabolism affecting (increasing) δ15N values in hair samples could only happen for a short period of time (weeks or several months, e.g. during pregnancy, starvation, tumor cachexia). It is not possible to maintain high δ15N levels due to health problems over a long time. The metabolic situation will come towards a steady state very soon that results in a “normal” or low δ15N level. I doubt that many Canadians are in a phase of bad health conditions. More likely, the most important factors for the variation of δ15N hair values are the different amounts and sources of dietary protein consumed by the individuals, and the kind of fertilizer used in agriculture. In general, δ15N values of poultry, beef or pigs are different because they receive feed based on specific components.

Page 28, line 523ff: Please consider my above mentioned comment about the influence of health situation on the δ15N hair values. δ13C values may be affected by certain diseases, e.g. diabetes or starvation, but to my knowledge, mostly the shift is low (< 1‰). I doubt that health and physiological conditions in the Canadian population are likely factors that strongly influence δ13C hair variation.

Page 30, line 563f: Examples for high-protein food items are all sources of animal protein: meat, fish, dairy products, eggs. Is there any reason to exclude fish, beef or pigs as a high-protein source for Cys? If I understood Nimni et al. 2007 [71] correctly, the ratio of Cys/Met in fish is lower than in meat from terrestrial sources, but fish and meat contain similar amounts of total S (from Met and Cys). Cys may also result from the breakdown of Met.

Page 31, line 584ff: Please add a reference supporting the argument that Mongolia imports a large proportion of its food from China that may have an influence on the δ34S hair values.

It may be possible that the extreme arid conditions and perhaps the occurrence of high δ34S values from evaporites or coal combustion may affect δ34S hair values at Mongolia. However, it is just an assumption.

CONCLUSION

Bases on my arguments above, there are some doubts about your statement that “Most of the non-dietary δ15N hair and δ13C hair probably relate to individual physiological and health variations within a population”. Individual physiology in metabolizing of food may affect the δ15N and δ13C hair values, but also the influence of individual features of food composition, which have the volunteers not been asked for in the FFQ, must not be neglected.

REFERENCES

Reference 33 is the same as reference 35.

Reference 64: the name of the first author (Nriagu NO) is written incorrectly.

Reviewer #2: I respect authors' substantial effort to collect the data including FFQ and isotopic compositions of scalp hairs, but find several issues regarding their interpretation of the results and mathematical modelings. Below I will point out my concerns one by one.

-major concerns-

Though I agree with authors on the potential utility of delta34S value for forensic applications, this is not the case for C and N isotopes. After all, the observed hair isotopic variation, particularly in delta15N, was not well-explained by the models in this study. Most of their discussions of possible factors that contributed the variation were just speculations without any scientific evidence. If authors want to discuss possible physiological/pathological controls on the hair delta15N variation, they should have designed the FFQ more suitably to meet their purpose from the beginning. Authors’ arguments to combine dietary surveys in several parts in the text (e.g., in abstract) and another argument in later sections (e.g., line426-435) that refer to the notorious inaccuracy of FFQ appears self-contradiction.

The statistical sections obviously need more explanation on the model selection processes, how authors overcame the multicollinearity, why they chose random forest out of many other machine learning procedures like SVM, etc.

-minor concerns-

Line 40: What about refereeing to the updated discussion for the delta15N discrimination inside animal bodies? (O’Connell, 2017)

Line 54-56: In regions like east Asia, peoples may have access to large amounts of sea foods regularly in supermarkets.

Line70: there are other studies that authors may as well refer to (e.g., Yoshinaga et al., 1996; Umezaki et al., 2016). Plus, they are encouraged to mention isotope studies dealing with finger nails too, though this is different body tissue (but same type protein, keratin) (e.g., Buchardt et al., 2007).

Line145-: this section ignores possible difference in the human hair growth phases, namely anagen/catagen/telogen.

Line214: authors should more explain the package VSURF to enable readers to understand what was going on during the data process on the variable selection.

Line 367: latitude/longitude data did not correlate with any environmental variables like MAP in Canada, though such environmental variables were not selected during model selections?

Line 400: there is no data for Japanese in Table 2.

Line400-402: I feel that the observed hair isotopic homogeneity for industrialized countries were caused by the mixing of isotopically distinct food items (vege, animal meat, dairy, fish, etc.) that might blind heterogeneity among local isotopic-baselines, rather than by a similarity in dietary habits.

Line426-: The credibility of FFQ depends on situations and how researchers design it (e.g., Hülsemann et al., 2017). I know that FFQ can be inaccurate because this approach relies on human memory and recording practices. Yet, saying just “noisy” sounds unprofessional.

Line 479-: In this section authors should mention that human scalp hairs are rapidly-growing body tissue.

Line 493: Is the authors’ argument here statistically correct?

Line 498-502: This part lacks scientific evidence.

Line 551: Is there no data for delta34S of agricultural crops in Canada?

Related papers

Buchardt, B., Bunch, V., Helin, P., 2007. Fingernails and diet: Stable isotope signatures of a marine hunting community from modern Uummannaq, North Greenland. Chemical Geology. 244, 316–329.

Hülsemann, F., Koehler, K., Wittsiepe, J., Wilhelm, M., Hilbig, A., Kersting, M., Braun, H., Flenker, U., Schänzer, W., 2017. Prediction of human dietary δ15N intake from standardised food records: validity and precision of single meal and 24-h diet data. Isotopes in Environmental and Health Studies. 53, 356–367.

O’Connell, T.C., 2017. ‘Trophic’ and ‘source’ amino acids in trophic estimation: a likely metabolic explanation. Oecologia. 184, 317–326.

Umezaki, M., Naito, Y.I., Tsutaya, T., Baba, J., Tadokoro, K., Odani, S., Morita, A., Natsuhara, K., Phuanukoonnon, S., Vengiau, G., Siba, P.M., Yoneda, M., 2016. Association between sex inequality in animal protein intake and economic development in the Papua New Guinea highlands: The carbon and nitrogen isotopic composition of scalp hair and fingernail. American Journal of Physical Anthropology. 159, 164–173.

Yoshinaga, J., Minagawa, M., Suzuki, T., Ohtsuka, R., Kawabe, T., Inaoka, T., Akimichi, T., 1996. Stable carbon and nitrogen isotopic composition of diet and hair of Gidra-speaking Papuans. American Journal of Physical Anthropology. 100, 23–34.

6. PLOS authors have the option to publish the peer review history of their article (what does this mean?). If published, this will include your full peer review and any attached files.

Reviewer #1: No

Reviewer #2: No

---

## [Author Response · Author response to Decision Letter 0]

17 Jun 2020

PONE-D-20-08053

Disentangling Dietary and Non-Dietary Controls of Hair Isotopic Variability in Human Populations: A case-study in Canada

PLOS ONE

Dear Dr. Bataille,

Thank you for submitting your manuscript to PLOS ONE. After careful consideration, we feel that it has merit but does not fully meet PLOS ONE’s publication criteria as it currently stands. Therefore, we invite you to submit a revised version of the manuscript that addresses the points raised during the review process.

We look forward to receiving your revised manuscript.

Kind regards,

Dorothée Drucker

Academic Editor

PLOS ONE

Additional Editor Comments:

Both reviewers underlined the importance of the data set presented in this paper. However, several points need to be thoroughly examined. The reliability and pertinence of the questionnaire should be further discussed and evaluated. 

We agree with the reviewer that our questionnaire had some issues. We have largely rewritten the discussion part on dietary choices to underline these limitations. We have refocus the main point of our manuscript on geographic variables by changing the title, abstract, introduction, and discussion. We have kept a section in the discussion on dietary choices but we now underline more clearly the limitations of our FFQ.

Some improvement about information category may be suggested in regard to the results and difficulties in the interpretation. 

We have abandoned the dietary vs. non dietary categories. We now speak of dietary choices, demographic variables and geographic/environmental variables. We also rewrote most of the introduction to clarify the different factor leading to isotopic variability.

The reviewer 1 raised an important point with the normalization of the isotopic results that has to be addressed by the authors, especially for the evaluation of possible inter-comparison with other studies on human hair. 

We appreciate the comment by reviewer 1. We have given a detailed response to this comment below (see reviewer 1) and by adding one reference in our text. We do not believe there is any normalization or comparison issues between our data and other labs. In fact, the Veizer Lab analyze more than 50,000 samples per year and regularly take part in inter-comparison lab procedures. 

The authors are also invited to consider additional bibliographic resources about human hair growth and isotopic recording. 

We have added all the suggested references

Discussion on the dietary vs non-dietary control on the 13C and 15N abundances in hair needs to be improved. 

We have largerly rewritten the discussion on dietary choices.

The hypothesis relating to physiological stress seems to be unlikely over a large period of time and geographical range. 

We have removed most of the discussion on physiological stress.

Elements are missing: data in Table 2 (see reviewer 2), information in the R script, (see reviewer 1). Pearson correlation coefficients in supplementary tables.

We have added new tables (table 3, 4 and 5) and figures (Fig. 3) in the main manuscript, we have updated the R script 

Journal Requirements:

Reviewers' comments:

Reviewer's Responses to Questions

Comments to the Author

1. Is the manuscript technically sound, and do the data support the conclusions?

Reviewer #1: Yes

Reviewer #2: Partly

2. Has the statistical analysis been performed appropriately and rigorously? 

Reviewer #1: N/A

Reviewer #2: I Don't Know

3. Have the authors made all data underlying the findings in their manuscript fully available?

Reviewer #1: No

Reviewer #2: Yes

4. Is the manuscript presented in an intelligible fashion and written in standard English?

Reviewer #1: Yes

Reviewer #2: Yes

5. Review Comments to the Author

Reviewer #1: This paper delivers data of δ13C, δ15N and δ34S values in hair samples from Canadian volunteers from all over the country. It contains isotopic information about self-reported composition of diet, different climates, soils and food components over Canada. At the periods of hair sampling the volunteers answered different questions, mainly about their dietary habits. Based on individual dietary and additional non-dietary information, the authors present all data. The data were statistically evaluated by machine-learning regression systems.

The author present important data, which were evaluated very well, and they present well-researched information. Furthermore, the authors developed excellent spatial distributions of hair isotope values.

After major revisions of the manuscript this paper should be published in PlosOne.

INTRODUCTION

Page 2, line 42:

In addition, δ15N values in diet and consequently in human tissues are influenced by the intake of legumes and the kind of fertilizer (organic vs. synthetic) used for food production, these factors should also be mentioned at that point.

We have rewritten this paragraph lines 71-78.

In Principle: How is the situation in Canada regarding soya? What’s about its consumption by the Canadians - directly from Soya products or via meat from animals, which were fed by soya?

We thank the reviewer for this idea, we have incorporated this point in the discussion lines 498-503.

Page 2, line 46:

The statement could be that δ34S values may also be influenced by trophic level. As I understood from the mentioned references, shifts of δ34S within the food web often are within the analytical uncertainties and the results are not always comprehensible (see also Tanz, N., & Schmidt, H. L. (2010). δ34S-value measurements in food origin assignments and sulfur isotope fractionations in plants and animals. Journal of agricultural and food chemistry, 58(5), 3139-3146.)

We modified the text and added the suggested reference lines 90-91

Page 3, line 62: The paper from Arneson and McAvoy 2005 deals with isotope fractionation from diet to mouse tissues. These results are not directly comparable with the dietary situation and metabolism of modern human adults. I my opinion it is not necessary to mention isotope fractionation from diet to humans at this point, therefore you may delete the sentence in the text.

We removed the sentence and citation as suggested

Page 3, line 64: There are some more references dealing with time-resolved incorporation of several isotopes into hair, which are worth mentioning:

• Huelsemann, F., Flenker, U., Koehler, K., & Schaenzer, W. (2009). Effect of a controlled dietary change on carbon and nitrogen stable isotope ratios of human hair. Rapid Communications in Mass Spectrometry: An International Journal Devoted to the Rapid Dissemination of Up‐to‐the‐Minute Research in Mass Spectrometry, 23(16), 2448-2454.

• Lehn, C., Lihl, C., & Roßmann, A. (2015). Change of geographical location from Germany (Bavaria) to USA (Arizona) and its effect on H–C–N–S stable isotopes in human hair. Isotopes in environmental and health studies, 51(1), 68-79.

• Lehn, C., Kalbhenn, E. M., Rossmann, A., & Graw, M. (2019). Revealing details of stays abroad by sequential stable isotope analyses along human hair strands. International journal of legal medicine, 133(3), 935-947.

We thank the reviewer for suggesting these references. We have added them where suggested, they were all very appropriate and useful to this paper.

Page 3, line 71-73: Both references “33” and “35” are identical.

We corrected as suggested.

Page 3, line 73 and page 4, line 75: Reference “35” is wrong at these points.

We replaced the citation line 75 and line 73 by more appropriate citation.

Page 5, line 101: Please check if citation “48” is right at that point.

We changed this reference as suggested. 

The following sentence: As mentioned above, for clarification all the “dietary” factors should be presented in a table.

We are not clear as to what the reviewer is referring to but we modified the entire manuscript removing the nomenclature of dietary vs. non dietary controls. We now speak about dietary choices, demographic and geographic/environmental controls on food isotopic baselines. This new wording should clarifiy the issues with dietary vs. non dietary.

Page 5 line 99ff: You may shorten the text by deletion of the whole part “Regions like Brazil ... .... not unique”.

We deleted as suggested.

Line 105ff: As mentioned above, if you want to utilize the terms “dietary” and “non-dietary”, there is a need for an exact definition. For example, “dietary”: depending on the composition of diet, and “non-dietary”: depending on agricultural production methods (and environmental/ climatic conditions?)

As mentioned by the reviewer, we removed the term dietary and non-dietary as they were confusing. We rewrote most of the introduction to account for these changes. 

MATERIALS AND METHODS

Page 8, line 173: As mentioned below, you may delete the sentence “C, N and S ...... isotopic fractionation [14].”

We deleted as suggested.

Page 8, line 182ff: It seems that for calibration of hair δ13C, δ15N, and δ34S values only inorganic standards were used. However, for analyzing hair samples keratin standards should be used for calibration. The authors mentioned they used in-house hair standards, but no isotope values are given. Why have you not taken USGS42 and USGS43 standards for calibration or at least for comparison with the results of your in house standards? In general, isotope values of hair samples analyzed by different groups or laboratories should be comparable, and this can only be achieved by the use of international isotope standards being of the same material (hair, at least keratin).

We thank the reviewer for this comment and we agree that data harmonization between laboratories is a key issue. Data harmonization is critical to the more than 50,000 samples analyzed every year in the Veizer Laboratory (https://isotope.uottawa.ca/en/about-us). 

We agree with the reviewer that comparing isotope data with standards of the same substrate is important, hence our comparison with the in-house hair internal standards. However, the only international standards for 13C are NBS-19 and LSVEC, (and LSVEC 13C value is now uncertain), all other materials are labelled as reference material (RM) and they do not define the vpdb scale. The USGS42-43 hair standards were created to fix the exchangeable issues with hair for d2H determination and the Veizer Lab contributed to that effort (see Meier-Augenstein W, Chartrand MM, Kemp HF, St-Jean G. An inter-laboratory comparative study into sample preparation for both reproducible and repeatable forensic 2H isotope analysis of human hair by continuous flow isotope ratio mass spectrometry. Rapid Commun Mass Spectrom. 2011;25(21):3331‐3338. doi:10.1002/rcm.5235). Unfortunately the spread of the two USGS hair standards is not great for d2H. In the case of N,C and S isotopes, the spread is even smaller and therefore cannot be used to normalized the data. In general for C and N, there are very little differences (if any) between inorganic and organic for the EA-IRMS. In our case, 3 out the 4 standards are organics caffeine(complex organic), L-glutamic acid (amino acid), Nicotiamide (vitamin)) calibrated versus USGS 40 and 41 (L-glutamic acids). The hair in-house stds have been defined years ago, before the USGS-42-43 came out. They have a much broader range of isotopic values for C, N, S and H and where compared to the USGS standards (see reference above).

Now sulfur isotope analysis is always a problem. Using similar matrix in this case can definitely result is better data mainly due to the oxygen issue. However, the spread of USGS42 vs 43 is only 2.7 permil. Additionally the RM USGS42 and 43 were calibrated using a normalized curve such as IAEA-S-1 gives a value of -0.3 permil (see USGS42 certificate). This is exactly what we do to our sequence, hence they are comparable to any data set. One can use IAEA-S-1 and S-2 directly with the sample runs like we did. In short, everything always comes back to S-1 (and S-2) normalized, regardless if one uses USGS42-43 since those two have also been defined with S-1 (and S-2). 

Page 9, Supplementary data:

In my download of the Plosone_SI.docx file, the Pearson correlation coefficients of Tables S3, S10 and S15 are not pictured.

We apologize about this. We have provided a file with updated figures. We have moved Table S3, S10 and S15 into the main manuscript.

Unfortunately I am not familiar with “R”, because I am using another statistical program (SPSS). I asked a colleague to open the R script. She found that “supl data called Data S1 does not include a sheet required for the script to run (‘Dietary’).”

The script was modified to run properly. We had modified the name of the excel sheet in S1 Data.

RESULTS

Table S6, S8 and S11 should be presented in the manuscript. Provinces in Table S11 should be named.

We re-added and combined most of the tables from the supplementary material back into the main manuscript and modified as suggested by the reviewer.

Chapter “Isotopic data in Canadian hair compared to other countries”

Page 15, line 292: The reference of Lehn et al. 2015 should be considered as well. It contains many δ13C, δ15N and δ34S data of worldwide collected human hair samples

• Lehn, C., Rossmann, A., & Graw, M. (2015). Provenancing of unidentified corpses by stable isotope techniques–presentation of case studies. Science & Justice, 55(1), 72-88).

We thank the reviewer for this suggestion. We have modified Table 2 to include this reference.

Page 16, line 323: incorrect spelling of δ34S

Corrected as suggested

For data evaluation the authors differentiate between “dietary” and “non-dietary” factors, but I could not find an exact definition thereof. It seems that “dietary” factors are based on individual information the volunteers replied to the questionnaires. However, the answers of the volunteers contain information about the composition of their diet, but also e.g. about smoking habits and possible hair dying. The authors should clarify the content or meaning of “dietary” factors used for data evaluations. This may avoid misunderstandings. For example, the proportion of C3 or C4 plants in diet could be considered as a “dietary” factor, but in the paper the proportion of C3 or C4 plants belongs to the “non-dietary” factors.

As mentioned above, we have clarified the vocabulary. We rewrote the title, abstract, introduction, and discussion separating dietary choices, demographic factors and geographic and environmental controls of isotopic variability in food systems. 

DISCUSSION

Page 23, line 401: “The overlapping of δ15N hair distribution between Canada, Europe and USA suggests a similar diet...” I do not agree to this statement, because “diet” or composition of diet in these regions is different. My suggestion: ... suggests in average the same amount of animal protein in diet and similar agricultural production methods for terrestrial food products between these countries.

We modified as suggested

Page 23, line 416: Change the sentence as follows: “The reasons for the low δ34S values in food-systems and human in Canada are likely related to...”

Furthermore, the kind of fertilizer may lead to low δ34S values. Which was the mostly applied fertilizer in Canada, and where did it come from?

The most common fertilizer is ammonium sulphates and come from the USA. It is likely that this fertilizer has a low d34S which reflect sulfur sources from crude oils and ore sulfides. We added a citation and a sentence l.424-425

Page 23, line 419: My suggestion: “... isotopically light S of anthropogenic sources (do you mean coal combustion?) from the eastern USA ...” Please check if this statement actually exists in the mentioned reference [45].

We modified the reference as suggested by the reviewer.

Page 24, line 433: It is very important to mention that self-reported diet is difficult to compare between the individuals. The personal specifications are not objective, but mostly relative. For example, I have the experience that people from the coast often underestimate their consumption of sea products, whereas people from regions in the inland, where it is rather unusual to eat plenty of sea products, overestimate its consumption.

In The FFQ some more personal details of diet are lacking: preference of organic or conventional food, amount of legumes in diet, preference of sea fish or freshwater fish.

We have added a new paragraph on FFQ and citation lines 434-444 citation and incorporated the reviewer comments.

Page 25, line 40: Please add some non-dietary factors that may explain δ15N hair variability, e.g. kind of fertilizer, agricultural production methods, amount of legumes in diet, intake of additional protein, health status.

We have tried to organize the paper to explain isotopic variability sequentially from active dietary choices (e.g., meat vs. no meat) to isotopic variability associated with local food isotopic baselines (e.g., dominant crop, fertilizer, climate). So we prefer to keep fertilizer and production methods for the later section of the discussion.

Page 25, line 449: What is about the intake of animal protein from dairy products in non-meat eaters?

We replaced animal protein by meat consumption.

Page 27, line 493: You may delete this sentence “Most .... unexplained.”

We removed as suggested

Just a brief comment: In accordance with your results relating to δ15N hair values, the results in Lehn et al. 2015 (Table 2) indicate that δ15N values in hair samples have the lowest potential (among C-N-S-H stable isotope values) for geographical discrimination into the different groups of origin. The statistical evaluations of the C-N-S-H stable isotopes in human hair samples were performed by canonical discriminant analyses.

We included this remark at the end of our d15N discussion (l.509-511)

Page 27, line 497: Please exchange “disease” by “certain diseases”.

We modified as suggested

Page 27, line 498ff: The following statement “the broad range of metabolism and health conditions between the individuals ... drive most of the δ15N hair variations” should be modified. Changes in metabolism affecting (increasing) δ15N values in hair samples could only happen for a short period of time (weeks or several months, e.g. during pregnancy, starvation, tumor cachexia). It is not possible to maintain high δ15N levels due to health problems over a long time. The metabolic situation will come towards a steady state very soon that results in a “normal” or low δ15N level. I doubt that many Canadians are in a phase of bad health conditions. More likely, the most important factors for the variation of δ15N hair values are the different amounts and sources of dietary protein consumed by the individuals, and the kind of fertilizer used in agriculture. In general, δ15N values of poultry, beef or pigs are different because they receive feed based on specific components.

We have removed most of the text concerning the transitional metabolic fractionation due to heath or catabolism throughout the manuscript. 

Page 28, line 523ff: Please consider my above mentioned comment about the influence of health situation on the δ15N hair values. δ13C values may be affected by certain diseases, e.g. diabetes or starvation, but to my knowledge, mostly the shift is low (< 1‰). I doubt that health and physiological conditions in the Canadian population are likely factors that strongly influence δ13C hair variation.

We have removed most of the text concerning the transitional metabolic fractionation due to heath or catabolism throughout the manuscript. 

Page 30, line 563f: Examples for high-protein food items are all sources of animal protein: meat, fish, dairy products, eggs. Is there any reason to exclude fish, beef or pigs as a high-protein source for Cys? If I understood Nimni et al. 2007 [71] correctly, the ratio of Cys/Met in fish is lower than in meat from terrestrial sources, but fish and meat contain similar amounts of total S (from Met and Cys). Cys may also result from the breakdown of Met.

We have modified this sentence as suggested. 

Page 31, line 584ff: Please add a reference supporting the argument that Mongolia imports a large proportion of its food from China that may have an influence on the δ34S hair values.

It may be possible that the extreme arid conditions and perhaps the occurrence of high δ34S values from evaporites or coal combustion may affect δ34S hair values at Mongolia. However, it is just an assumption.

We have modified this sentence to avoid confusion but we added a statement about other countries with low d34S in human hair l.592.

CONCLUSION

Bases on my arguments above, there are some doubts about your statement that “Most of the non-dietary δ15N hair and δ13C hair probably relate to individual physiological and health variations within a population”. Individual physiology in metabolizing of food may affect the δ15N and δ13C hair values, but also the influence of individual features of food composition, which have the volunteers not been asked for in the FFQ, must not be neglected.

We have modified our conclusion to take into account the reviewer’s comments.

REFERENCES

Reference 33 is the same as reference 35.

Reference 64: the name of the first author (Nriagu NO) is written incorrectly.

We corrected both references. We also cleaned all other references adding volume, issues and correcting typos when necessary. 

Reviewer #2: I respect authors' substantial effort to collect the data including FFQ and isotopic compositions of scalp hairs, but find several issues regarding their interpretation of the results and mathematical modelings. Below I will point out my concerns one by one.

-major concerns-

Though I agree with authors on the potential utility of delta34S value for forensic applications, this is not the case for C and N isotopes. After all, the observed hair isotopic variation, particularly in delta15N, was not well-explained by the models in this study. Most of their discussions of possible factors that contributed the variation were just speculations without any scientific evidence. If authors want to discuss possible physiological/pathological controls on the hair delta15N variation, they should have designed the FFQ more suitably to meet their purpose from the beginning. Authors’ arguments to combine dietary surveys in several parts in the text (e.g., in abstract) and another argument in later sections (e.g., line426-435) that refer to the notorious inaccuracy of FFQ appears self-contradiction.

We agree with the reviewer. We agree that the main novelty of our paper are the d34S intepretations. We also agree that the FFQ data has some important limitations. We initially debated about writing a paper only about d34S variability. However, we felt that the d15N, d13C and FFQ data had some value and should be published along with their limited interpretations. As the reviewer can see, we have rewritten a large portion of the title, abstract, introduction, result and discussion sections to put more emphasis on the geographic and environmental controls. In the section discussing the influence of dietary choices on isotopic variability, we emphasized the limitations of the FFQ. We believe that refocusing the manuscript towards geographic and environmental controls of local food isotopic baselines will help underline the value of this work. 

The statistical sections obviously need more explanation on the model selection processes, how authors overcame the multicollinearity, why they chose random forest out of many other machine learning procedures like SVM, etc.

We have rewritten and added a new paragraph to explain: 1) What is random forest (l.203-211), 2) why we choose random forest against other algorithms (l.199-203) and 3) how we overcame multicolinearity using the VSURF (l.211- 214)

-minor concerns-

Line 40: What about refereeing to the updated discussion for the delta15N discrimination inside animal bodies? (O’Connell, 2017)

We added as suggested.

Line 54-56: In regions like east Asia, peoples may have access to large amounts of sea foods regularly in supermarkets.

We modified the sentence. 

Line70: there are other studies that authors may as well refer to (e.g., Yoshinaga et al., 1996; Umezaki et al., 2016). Plus, they are encouraged to mention isotope studies dealing with finger nails too, though this is different body tissue (but same type protein, keratin) (e.g., Buchardt et al., 2007).

We added both of the indicated references.

Line145-: this section ignores possible difference in the human hair growth phases, namely anagen/catagen/telogen.

We added a sentence and a citation to underline this uncertainty l.150-154

Line214: authors should more explain the package VSURF to enable readers to understand what was going on during the data process on the variable selection.

We added more details about the VSURF algorithm (l.211- 214)

Line 367: latitude/longitude data did not correlate with any environmental variables like MAP in Canada, though such environmental variables were not selected during model selections?

We are not sure what the reviewer is asking with this comment. Only non-redundant predictors are selected within the model. Latitude and longitude do correlate with MAP but in the final model only a variable that adds to the predictive power (reducing out-of-bag error of the model) is selected by VSURF. Table 6 presents only the significant predictors selected by VSURF for each isotopic system and each series of variables. We have also added Fig. 3 that summarizes the correlations between isotope data and other covariates. 

Line 400: there is no data for Japanese in Table 2.

We have modified table 2 to follow advices from reviewer 1 and 2. We removed Japan from that sentence.

Line400-402: I feel that the observed hair isotopic homogeneity for industrialized countries were caused by the mixing of isotopically distinct food items (vege, animal meat, dairy, fish, etc.) that might blind heterogeneity among local isotopic-baselines, rather than by a similarity in dietary habits.

We modified the sentence to take the reviewer comment into account.

Line426-: The credibility of FFQ depends on situations and how researchers design it (e.g., Hülsemann et al., 2017). I know that FFQ can be inaccurate because this approach relies on human memory and recording practices. Yet, saying just “noisy” sounds unprofessional.

We thank the reviewer for this comment we rewrote this section (l.434-444)

Line 479-: In this section authors should mention that human scalp hairs are rapidly-growing body tissue.

We added the sentence line 613

Line 493: Is the authors’ argument here statistically correct?

We removed that sentence.

Line 498-502: This part lacks scientific evidence.

As suggested by reviewer 1 and 2, we have removed all our mention of health and metabolic fractionation as we have no evidences for these claims.

Line 551: Is there no data for delta34S of agricultural crops in Canada?

To our knowledge we have cited all the papers that discuss soil and plant d34S in Canada.

Related papers

Buchardt, B., Bunch, V., Helin, P., 2007. Fingernails and diet: Stable isotope signatures of a marine hunting community from modern Uummannaq, North Greenland. Chemical Geology. 244, 316–329.

Hülsemann, F., Koehler, K., Wittsiepe, J., Wilhelm, M., Hilbig, A., Kersting, M., Braun, H., Flenker, U., Schänzer, W., 2017. Prediction of human dietary δ15N intake from standardised food records: validity and precision of single meal and 24-h diet data. Isotopes in Environmental and Health Studies. 53, 356–367.

O’Connell, T.C., 2017. ‘Trophic’ and ‘source’ amino acids in trophic estimation: a likely metabolic explanation. Oecologia. 184, 317–326.

Umezaki, M., Naito, Y.I., Tsutaya, T., Baba, J., Tadokoro, K., Odani, S., Morita, A., Natsuhara, K., Phuanukoonnon, S., Vengiau, G., Siba, P.M., Yoneda, M., 2016. Association between sex inequality in animal protein intake and economic development in the Papua New Guinea highlands: The carbon and nitrogen isotopic composition of scalp hair and fingernail. American Journal of Physical Anthropology. 159, 164–173.

Yoshinaga, J., Minagawa, M., Suzuki, T., Ohtsuka, R., Kawabe, T., Inaoka, T., Akimichi, T., 1996. Stable carbon and nitrogen isotopic composition of diet and hair of Gidra-speaking Papuans. American Journal of Physical Anthropology. 100, 23–34.

We have added all these references in our manuscript.

6. PLOS authors have the option to publish the peer review history of their article (what does this mean?). If published, this will include your full peer review and any attached files.

Do you want your identity to be public for this peer review? For information about this choice, including consent withdrawal, please see our Privacy Policy.

Reviewer #1: No

Reviewer #2: No

---

## [Decision Letter · Decision Letter 1]

7 Jul 2020

PONE-D-20-08053R1

Assessing Geographic Controls of Hair Isotopic Variability in Human Populations: A case-study in Canada

PLOS ONE

Dear Dr. Bataille,

Thank you for submitting your manuscript to PLOS ONE. After careful consideration, we feel that it has merit but does not fully meet PLOS ONE’s publication criteria as it currently stands. Therefore, we invite you to submit a revised version of the manuscript that addresses the points raised during the review process.

Both reviewers and myself agree that you responded to the critics in a satisfactory way. As a result, the manuscript has been thoroughly revised and the structure wisely re-organised. Only minor revisions are necessary to make the paper ready for publication.

We look forward to receiving your revised manuscript.

Kind regards,

Dorothée Drucker

Academic Editor

PLOS ONE

Reviewers' comments:

Reviewer's Responses to Questions

**Comments to the Author**

1. If the authors have adequately addressed your comments raised in a previous round of review and you feel that this manuscript is now acceptable for publication, you may indicate that here to bypass the “Comments to the Author” section, enter your conflict of interest statement in the “Confidential to Editor” section, and submit your "Accept" recommendation.

Reviewer #1: All comments have been addressed

Reviewer #2: All comments have been addressed

2. Is the manuscript technically sound, and do the data support the conclusions?

Reviewer #1: Yes

Reviewer #2: Yes

3. Has the statistical analysis been performed appropriately and rigorously? 

Reviewer #1: Yes

Reviewer #2: Yes

4. Have the authors made all data underlying the findings in their manuscript fully available?

Reviewer #1: Yes

Reviewer #2: Yes

5. Is the manuscript presented in an intelligible fashion and written in standard English?

Reviewer #1: Yes

Reviewer #2: Yes

6. Review Comments to the Author

Reviewer #1: First of all, my compliments to the authors for the highly improved quality of the revised manuscript!

From my point of view, there are only some minor changes or corrections needed:

ABSTRACT

Line 14: At the end of the sentence, I would add „of a current population“, because bones and teeth are also useful to investigate human nutrition, but not on living humans.

INTRODUCTION

Line 48: My suggestion for a change: “In many countries … becoming increasingly homogeneous due to the globalization….”, and “On the other hand, regional dietary traditions….. may contribute to a higher isotopic variability….” For example, also in less industrialized regions of India, Central America, Southern America or Africa available foods are influenced by globalization, particular in urban regions, but in more rural regions, traditional diets are still common.

MATERIAL AND METHODS

I am still convinced that stable isotope analyses of the international hair standards USGS 42/USGS 43 would have been useful to make sure that the values in your hair samples (especially for d34S) are comparable with those of the other laboratories.

Line 164: Please correct the sentence: “of” is missing at a certain position of the sentence, and “consumer”

RESULTS

TABLE 2: The isotope values given in all the tables should be consistent (two decimal places). Please add a negative sign to all d13C values, and correct the sample size of Asian hair samples of [70], it should be 137.

Line 332: Please correct “d34N”.

Line 345: Please check your statement regarding the countries with “higher d15N hair values”.

DISCUSSION

Line 400ff: As I have already mentioned before, I do not completely agree with your statement that “Canada shows a similar trend to that observed in other industrialized countries such as Europe and the USA” due to homogenization of diet. Different to US or Canada, Europa consists of a lot of small countries with different historical and cultural realities that are also reflected in diet; therefore a relatively high variability of stable isotope values in hair samples from the different countries exists (see Hülsemann et al. 2015, Valenzuela et al. 2012).

Line 416: I assume that you mean reference [70] instead of [6].

Line 616ff: To get an easier understanding of your statements, the term “individuals” should be exchanges by “Participant + its number”. Do you mean that Participant 4 “stopped eating dairy products for a prolonged period” in the near past? You may add that timeframe to clarify that this is visible in the most recent hair samples. Furthermore, I do not like the term “anomalously” in that context. In my opinion, “variable” is sufficient.

Line 623: I would say that “d34S hair values may be dominantly controlled by the geography (or geographical origin) of the diet”.

CONCLUSION

Line 636: What do you mean by “temporary stable patterns”? I would suggest to delete the two words and add “food systems across modern (or contemporary) Canada transmitted to……

Line 639: My suggestion would be: “Our work also paves the way for (several) promising applications of S isotopes in food and forensic science.” d34S values in food as well as in human hair or collagen samples are already being used for food traceability and for provenancing of unknown individuals in forensics.

Reviewer #2: The ms has been improved. There are now only a few trivial concerns I can raise.

-line154 and some other places: "proprieties" should be "properties"?

-line866: "fractionation microbial processes" should be "fractionation by microbial processes"?

-There are two Acknowledgment sections in the revised text.

-several tables and figures; is it necessary to show hundredths-place digits?

7. PLOS authors have the option to publish the peer review history of their article (what does this mean?). If published, this will include your full peer review and any attached files.

Reviewer #1: No

Reviewer #2: No

---

## [Author Response · Author response to Decision Letter 1]

17 Jul 2020

PONE-D-20-08053R1

Assessing Geographic Controls of Hair Isotopic Variability in Human Populations: A case-study in Canada

PLOS ONE

Dear Dr. Bataille,

Thank you for submitting your manuscript to PLOS ONE. After careful consideration, we feel that it has merit but does not fully meet PLOS ONE’s publication criteria as it currently stands. Therefore, we invite you to submit a revised version of the manuscript that addresses the points raised during the review process.

Both reviewers and myself agree that you responded to the critics in a satisfactory way. As a result, the manuscript has been thoroughly revised and the structure wisely re-organised. Only minor revisions are necessary to make the paper ready for publication.

Thank you for your suggestions. We greatly appreciated this review process.

We look forward to receiving your revised manuscript.

Kind regards,

Dorothée Drucker

Academic Editor

PLOS ONE

Reviewers' comments:

Reviewer's Responses to Questions

Comments to the Author

1. If the authors have adequately addressed your comments raised in a previous round of review and you feel that this manuscript is now acceptable for publication, you may indicate that here to bypass the “Comments to the Author” section, enter your conflict of interest statement in the “Confidential to Editor” section, and submit your "Accept" recommendation.

Reviewer #1: All comments have been addressed

Reviewer #2: All comments have been addressed

2. Is the manuscript technically sound, and do the data support the conclusions?

Reviewer #1: Yes

Reviewer #2: Yes

3. Has the statistical analysis been performed appropriately and rigorously? 

Reviewer #1: Yes

Reviewer #2: Yes

4. Have the authors made all data underlying the findings in their manuscript fully available?

Reviewer #1: Yes

Reviewer #2: Yes

5. Is the manuscript presented in an intelligible fashion and written in standard English?

Reviewer #1: Yes

Reviewer #2: Yes

6. Review Comments to the Author

Reviewer #1: First of all, my compliments to the authors for the highly improved quality of the revised manuscript!

From my point of view, there are only some minor changes or corrections needed:

We wanted to thank the reviewer for the thorough and details comments that have greatly improved this manuscript

ABSTRACT

Line 14: At the end of the sentence, I would add „of a current population“, because bones and teeth are also useful to investigate human nutrition, but not on living humans.

We corrected as suggested

INTRODUCTION

Line 48: My suggestion for a change: “In many countries … becoming increasingly homogeneous due to the globalization….”, and “On the other hand, regional dietary traditions….. may contribute to a higher isotopic variability….” For example, also in less industrialized regions of India, Central America, Southern America or Africa available foods are influenced by globalization, particular in urban regions, but in more rural regions, traditional diets are still common.

We corrected as suggested

MATERIAL AND METHODS

I am still convinced that stable isotope analyses of the international hair standards USGS 42/USGS 43 would have been useful to make sure that the values in your hair samples (especially for d34S) are comparable with those of the other laboratories.

This is a good point. We have placed an order for these standards but they are not cheap…

Line 164: Please correct the sentence: “of” is missing at a certain position of the sentence, and “consumer”

We modified the sentence

RESULTS

TABLE 2: The isotope values given in all the tables should be consistent (two decimal places). Please add a negative sign to all d13C values, and correct the sample size of Asian hair samples of [70], it should be 137.

We corrected as suggested

Line 332: Please correct “d34N”.

We corrected as suggested

Line 345: Please check your statement regarding the countries with “higher d15N hair values”.

We corrected the statement

DISCUSSION

Line 400ff: As I have already mentioned before, I do not completely agree with your statement that “Canada shows a similar trend to that observed in other industrialized countries such as Europe and the USA” due to homogenization of diet. Different to US or Canada, Europa consists of a lot of small countries with different historical and cultural realities that are also reflected in diet; therefore a relatively high variability of stable isotope values in hair samples from the different countries exists (see Hülsemann et al. 2015, Valenzuela et al. 2012).

We modified the language to account for this comment

Line 416: I assume that you mean reference [70] instead of [6].

Yes we modified as suggested.

Line 616ff: To get an easier understanding of your statements, the term “individuals” should be exchanges by “Participant + its number”. Do you mean that Participant 4 “stopped eating dairy products for a prolonged period” in the near past? You may add that timeframe to clarify that this is visible in the most recent hair samples. Furthermore, I do not like the term “anomalously” in that context. In my opinion, “variable” is sufficient.

We replaced the term individual by participant throughout the manuscript. We also specified participant number for each individual in this section.

Line 623: I would say that “d34S hair values may be dominantly controlled by the geography (or geographical origin) of the diet”.

We changed as suggested.

CONCLUSION

Line 636: What do you mean by “temporary stable patterns”? I would suggest to delete the two words and add “food systems across modern (or contemporary) Canada transmitted to……

We changed as suggested.

Line 639: My suggestion would be: “Our work also paves the way for (several) promising applications of S isotopes in food and forensic science.” d34S values in food as well as in human hair or collagen samples are already being used for food traceability and for provenancing of unknown individuals in forensics.

Reviewer #2: The ms has been improved. There are now only a few trivial concerns I can raise.

-line154 and some other places: "proprieties" should be "properties"?

We changed as suggested throughout the manuscript.

-line866: "fractionation microbial processes" should be "fractionation by microbial processes"?

We added “by”

-There are two Acknowledgment sections in the revised text.

We corrected as suggested.

-several tables and figures; is it necessary to show hundredths-place digits?

We changed most tables to 1 significant digit except for table 7 for which the second significant digit is important particularly in the standard deviation.

7. PLOS authors have the option to publish the peer review history of their article (what does this mean?). If published, this will include your full peer review and any attached files.

Do you want your identity to be public for this peer review? For information about this choice, including consent withdrawal, please see our Privacy Policy.

Reviewer #1: No

Reviewer #2: No

---

## [Editor Report · Decision Letter 2]

21 Jul 2020

Assessing Geographic Controls of Hair Isotopic Variability in Human Populations: A case-study in Canada

PONE-D-20-08053R2

Dear Dr. Bataille,

We’re pleased to inform you that your manuscript has been judged scientifically suitable for publication and will be formally accepted for publication once it meets all outstanding technical requirements.

Kind regards,

Dorothée Drucker

Academic Editor

PLOS ONE
---

## [Editor Report · Acceptance letter]

24 Jul 2020

PONE-D-20-08053R2 

Assessing Geographic Controls of Hair Isotopic Variability in Human Populations: A case-study in Canada 

Dear Dr. Bataille:

I'm pleased to inform you that your manuscript has been deemed suitable for publication in PLOS ONE. Congratulations! Your manuscript is now with our production department. 

Kind regards, 

on behalf of

Dr. Dorothée Drucker 

Academic Editor

PLOS ONE